

# Probabilistic forecasting of plausible debris flows from Nevado de Colima (México) using data from the Atenquique debris flow, 1955

Andrea Bevilacqua[1,2], Abani K. Patra[3,2], Marcus I. Bursik[1], E. Bruce Pitman[4], José Luis Macías[5], Ricardo Saucedo[6], and David Hyman[1]

[1]*Department of Earth Sciences, SUNY at Buffalo, NY, 14260*
[2]*Computational Data Science and Engineering program, SUNY at Buffalo, NY, 14260*
[3]*Department of Mechanical and Aerospace Engineering, SUNY at Buffalo, NY, 14260*
[4]*Department of Materials Design and Innovation, SUNY at Buffalo, NY, 14260*
[5]*Departamento de Vulcanología, Instituto de Geofísica, UNAM, DF, 04510*
[6]*Instituto de Geología, Facultad de Ingeniería, UASLP, SLP, 78240*

**Correspondence:** Andrea Bevilacqua (bev87@hotmail.it)

**Abstract.** We detail a new prediction-oriented procedure aimed at volcanic hazard assessment based on geophysical mass flow models constrained with heterogeneous and poorly defined data. Our method relies on an itemized application of the empirical falsification principle over an arbitrarily wide envelope of possible input conditions. We thus provide a first step towards a objective and partially automated experimental design construction. In particular, instead of fully calibrating model

inputs on past observations, we create and explore more general requirements of consistency, and then we separately use each piece of empirical data to remove those input values that are not compatible with it, hence defining partial solutions to the inverse problem. This has several advantages compared to a traditionally posed inverse problem: (i) the potentially non-empty inverse images of partial solutions of multiple possible forward models characterize the solutions to the inverse problem; (ii) the partial solutions can provide hazard estimates under weaker constraints, potentially including extreme cases

that are important for hazard analysis; (iii) if multiple models are applicable, specific performance scores against each piece of empirical information can be calculated. We apply our procedure to the case study of the Atenquique volcaniclastic debris flow, which occurred on the flanks of Nevado de Colima volcano (México), 1955. We adopt and compare three depth averaged models currently implemented in the TITAN2D solver, available from vhub.org. The associated inverse problem is not well-posed if approached in a traditional way. We show that our procedure can extract valuable information for hazard assessment,

allowing the exploration of the impact of synthetic flows similar to those that occurred in the past, but different in plausible ways. The implementation of multiple models is thus a crucial aspect of our approach, as they can allow the covering of other plausible flows. We also observe that model selection is inherently linked to the inversion problem.





# 1 Introduction

Hazard assessment of geophysical mass flows, such as landslides or pyroclastic flows, usually relies on the reconstruction of past flows that occurred in the region of interest. The available pieces of data $D_i \in \mathcal{D}$, are commonly related to the properties of the deposit left by the flows, and to historical documentation. In general, this information can be affected by relevant sources of uncertainty (e.g., erosion and re-mobilization, superposition of subsequent events, unknown duration and source). Physical models provide us with a deterministic system to relate inputs and outputs of the dynamical system of the mass flow (Gilbert, 1991; Patra et al., 2018a) and are traditionally used to explore "what-if" scenarios and more recently in ways to supplement observed data with physically consistent synthetic data (Bayarri et al., 2015) for probabilistic analysis.

In a probabilistic framework, for each model $M \in \mathcal{M}$ we define $(M, P_M)$, where $P_M$ is a probability measure over the parameter space of $M$. While the support of $P_M$ can be restricted to a single value by solving an inverse problem for the optimal reconstruction of a particular flow, the inverse problem is not always well-posed (Tarantola and Valette, 1982; Tarantola, 1987). That is, no input data, or multiple input data, are able to produce outputs consistent with the observed information. Sometimes the strict replication of a past flow may lead to overconstraining the model, especially if we are interested in the general predictive capabilities of a model over a whole range of possible future events. In this study, we generalize a poorly constrained inverse problem, decomposing it into a hierarchy of simpler problems. Our purpose is to provide a new prediction-oriented formulation for use in hazard assessment problems.

## 1.1 Probabilistic description

Our approach is characterized by three steps:

1. For each model $M_j \in \mathcal{M}$, we initially set up a uniformly probabilized *general input space* that we define as arbitrarily wide, namely $(\Omega_0^j, \mathcal{F}_0^j, P_0^j)$. At this stage, we only require essential properties of feasibility in the models, namely the existence of the numerical output and the realism of the underlying physics.

2. After a preliminary screening, we characterize a *specialized input space* $\Omega^j \subseteq \Omega_0^j$, under additional requirements of plausibility [1] that are related to the macroscopic properties of the outputs. For instance, a robust numerical simulation without spurious effects, meaningful flow dynamics, and/or the capability to inundate a designated region.

3. Through more detailed testing, $\forall i \in I$, we can thus define the subspace $\Omega_i^j \subseteq \Omega^j$ of the inputs that are consistent with a piece of empirical data $D_i$.

We remark that the specialized input space is the inverse image of the set of plausible outputs $D_G$ through the input-output functions $f_j$ characterizing the models (see following section). We also note that, $\forall j$, all the described subspaces of $\Omega_0^j$, if not negligible, are trivially probabilized by the measures $P^j$ and $P_i^j$, defined by:

$$P_i^j(A) = P_0^j(A)/P_0^j(\Omega_i^j),$$

---

[1] our notion of *output plausibility* is not related to the *model plausibility* defined in (Farrell et al., 2015)




for each measurable set $A \in \mathcal{F}_i^j := \left\{ \Omega_i^j \cap B : \ B \in \mathcal{F}_0^j \right\}$.

The philosophy of our method is based on an itemized application of the empirical falsification principle of K.R. Popper (Popper, 1959), over an arbitrarily wide envelope of possible input conditions. The construction of the subspaces $(\Omega_i^j)_{i \geq 1}$ has several advantages compared to a traditionally posed inverse problem:

 – the intersection space $\Theta^j := \bigcap_i \Omega_i^j$ describes the set of the inputs that solves the inverse problem;

 – the partial solutions $(\Omega_i^j)_{i \geq 1}$ provide information concerning flows that partially solve the inverse problem, and they may exist even if $\Theta^j = \emptyset$;

 – each probability $P^j(\Omega_i^j)$ represents a performance score of the adopted model against the piece of empirical data $D_i$, and can therefore be used for model selection purposes.

10 ## 1.2 Functional structures

Our meta-modeling framework is fully described in Figure 1. Let us assume that each model $M_j \in \mathcal{M}$ is represented by an operator:

$$f_{M_j} : \Omega_0^j \longrightarrow \mathbb{R}^d,$$

where $d \in \mathbb{N}$ is a dimensional parameter which is independent on the model chosen, and characterizes a common output space. We also define the global set of feasible inputs:

$$\Omega_G := \bigsqcup_j \Omega_0^j.$$

This puts all the models in a natural meta-modeling framework. Then, we characterize the codomain $D_G \subset \mathbb{R}^d$ of plausible outputs, that is the target of our simulations - it includes all the outputs consistent with the observed data, plus additional outputs which differ in arbitrary, but plausible ways. For the sake of simplicity, we use the same notation for each piece of data $D_i$ and the set of outputs consistent with it. Then $\forall j$, the specialized input space is defined by:

$$\Omega^j = f_{M_j}^{-1}(D_G).$$

In a similar way, $\forall i, \Omega_i^j = f_{M_j}^{-1}(D_i)$, and for this reason those sets are called partial solutions to the inverse problem.

We remark that the implementation of multiple models is a crucial aspect in our approach. Typically, the available models are not able to entirely cover $D_G$, and:

$$D_G \setminus \bigcup_j f_{M_j}\left(\Omega_0^j\right) \neq \emptyset.$$

The investigation of $D_G$ and the quality of the solutions will clearly improve as we add more models based on new knowledge of the underlying processes, especially when there is uncertainty as to which model is best in a particular situation. We show that the solution of the partial inverse problems and model selection are strongly linked.





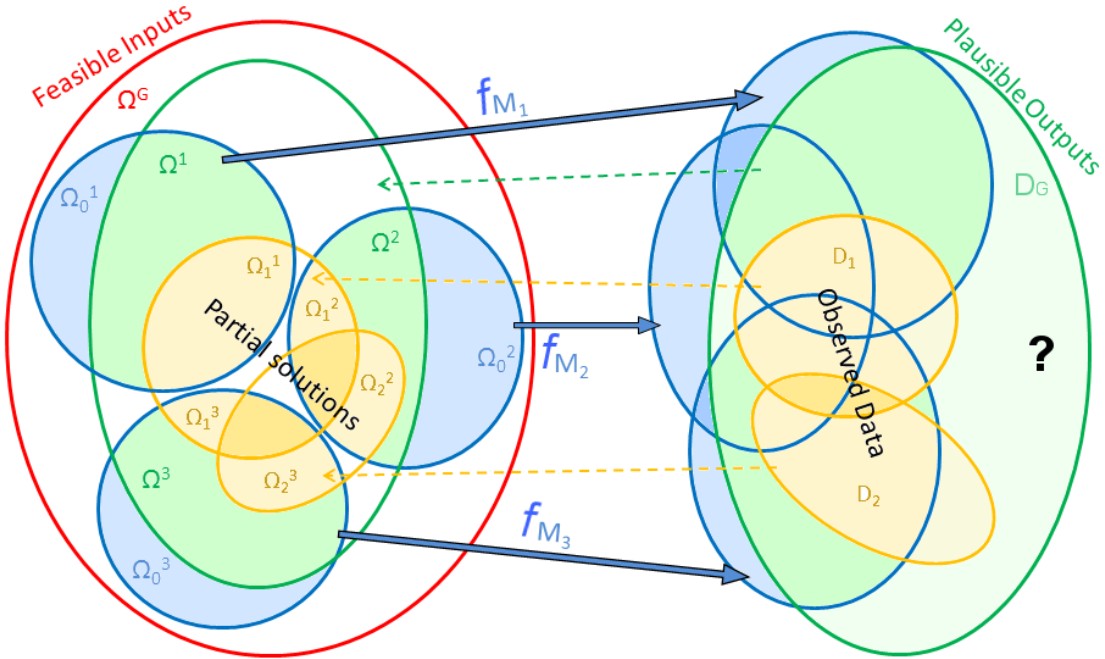

**Figure 1.** Diagram of input spaces, model functions, and output space (blue), with feasible inputs domain (red), plausible output codomain and specialized inputs (green), and observed data and partial solutions subsets (orange). The question mark emphasizes that the covering of other plausible outputs could be enabled, adding more models if necessary.

### 1.3 Geophysical case study

We apply our procedure to the case study of the Atenquique volcaniclastic debris flow, which occurred on the flanks of Nevado de Colima volcano (México) in 1955. We adopt and compare the three depth averaged models *Mohr-Coulomb* (MC) (Savage and Hutter, 1989), *Pouliquen-Forterre* (PF) (Pouliquen, 1999; Forterre and Pouliquen, 2002; Pouliquen and Forterre, 2002) and *Voellmy-Salm* (VS) (Voellmy, 1955; Salm et al., 1990), based on the Saint-Venant equations. Input spaces are explored by Monte Carlo simulation based on Latin Hypercube sampling (McKay et al., 1979; Owen, 1992b; Stein, 1987; Ranjan and Spencer, 2014; Ai et al., 2016). The three models are incorporated in our large scale mass flow simulation framework TITAN2D (Patra et al., 2005, 2006; Yu et al., 2009; Aghakhani et al., 2016; Patra et al., 2018b). So far, TITAN2D has been successfully applied to the simulation of different geophysical mass flows with specific characteristics (Sheridan et al., 2005; Rupp et al., 2006; Norini et al., 2009; Charbonnier and Gertisser, 2009; Procter et al., 2010; Sheridan et al., 2010; Sulpizio et al., 2010; Capra et al., 2011). Several studies involving TITAN2D were also directed towards statistical study of geophysical flows, focusing on uncertainty quantification (Dalbey et al., 2008; Dalbey, 2009; Stefanescu et al., 2012a, b), or on the more efficient production of hazard maps (Bayarri et al., 2009; Spiller et al., 2014; Bayarri et al., 2015; Ogburn et al., 2016).

The rest of the study is organized as follows. In section 2, we present our case study - debris flows from Nevado de Colima; in section 3, we introduce the physical models, we define and parameterize their input spaces, and we design our Monte Carlo





simulation; in section 4 we statistically describe the characteristics of the outputs and contributing variables, mapping them globally and detailing them locally. Finally, in section 5, we use multiple pieces of information regarding an historical debris flow to condition the input space, and then we compare the performance of the models over that space. The results show that model selection is inherently linked to the inversion problem, that is, model performance depends on which of the observed

data we seek to reproduce. This is a fundamental aspect to consider in the development of multi-model solvers, dynamically selecting the model based on performance against local data.

## 2   Nevado de Colima volcano and Barranca de Atenquique

The Colima Volcanic Complex is located in the western portion of the Trans-Mexican Volcanic Belt (small box in Fig. 2). It consists of a N–S volcanic chain formed by Cántaro, Nevado de Colima, and Colima volcanoes, within the Colima Graben

(Luhr and Carmichael, 1990). It began forming 1.7 Ma ago with the growth of Cántaro Volcano (2800 m.a.s.l), and continued with the formation of Nevado de Colima volcano 0.53 Ma ago (Allan, 1986; Robin et al., 1987). Activity was intermittent during the Pleistocene and Holocene and continues today at Colima volcano (3860 m.a.s.l.), south of Nevado (Saucedo et al., 2010; Zobin et al., 2015; Macorps et al., 2018).

Nevado de Colima (4320 m.a.s.l.) occupies the central part of the volcanic complex, being the most voluminous of the three

volcanoes (300–400 km$^3$). It is characterized by three or four horseshoe-shaped craters (Cortés et al., 2010). The youngest crater is 4 km wide and opens to the east with 100 m deep, vertical walls. This structure contains the summit Picacho dome (4300 m). The eastern flank of Nevado exposes a thick sequence of debris flow, fluvial, pyroclastic flow and debris avalanche deposits known as the Atenquique Formation (Mooser, 1961; Cortés et al., 2005; Saucedo et al., 2008), covered by younger deposits. The crater morphology directs a large part of the drainage from the volcano into the Atenquique ravine, located on

the ENE flank of the volcano and ∼25 km long. Figure 2 shows the ravine and its surrounding topography.

Plot (2b) includes elevation isolines from the SRTM DEM of 30-m cell size for UTM Zone 13N (NASA (JPL), 2014). The drainage begins at an elevation of 4000 m on the eastern flank of Nevado, is occupied by the perennial Atenquique river, and ends at its junction with the perennial Tuxpan River at 1040 m. Between, 4000 and 2000 m in altitude, the ravine has an average slope of 34°. At an elevation of 1800 m, the Atenquique ravine is cut off by a 250 m cliff in the vicinity of the junction with the

Dos Volcanes dry ravine. Beyond this cliff, the slope of the ravine suddenly decreases from 34° to 18°, keeping this gradient to an elevation of 1600 m at 11.5 km from the summit. Between this site and the catchments of the Los Plátanos and Arroyo Seco dry ravines, located at ∼21.5 km at an elevation of 1120 m, the ravine gradient varies from 10° to 6°. The town of Atenquique is located at an altitude of 1050 m, and 18.5 km (25 km along the ravine path) from the head of the ravine. There, the ravine channel has a gradient of 5°, which is maintained down to the confluence with the Tuxpan River. Atenquique ravine has an

average width of 30 m, although in the vicinity of Atenquique village it is up to 200 m wide.





# BARRANCA DE ATENQUIQUE

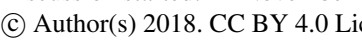

**Figure 2.** Barranca de Atenquique (México) overview. (a) Stratigraphic sections of (Saucedo et al., 2008) are marked with red dots, including 5 preferred locations (stars) and major ravines. Shaded sites are detailed in Supporting Information. Initial source piles are marked by blue dots. Coordinates and projection are UTM zone 13N WGS84. (b) Digital elevation map including isolines (NASA (JPL), 2014). Volume partition percentages among sources are reported. Regional map is included in a small box.

## 2.1 The Atenquique volcaniclastic debris flow, 1955

On 16 October, 1955, at 10:45 am, the inhabitants of Atenquique were surprised by the sudden arrival of an 8–9-m high wave carrying mud, boulders and tree trunks that devastated the buildings in the town and four bridges, including the railroad bridge. More than 23 people died, and the flood leveled everything but the tower of the church and the upper part of the market place





that luckily served as shelter for survivors (Ponce-Segura, 1983; Saucedo et al., 2008). During the peak flow, eyewitnesses observed that complete walls of buildings were displaced several meters by the flood prior to their collapse. The deposits are exposed along the Atenquique, Arroyo Seco, Los Plátanos and Dos Volcanes ravines, and their distribution, stratigraphy, granulometry, and volume have been described in Saucedo et al. (2008). Sixty stratigraphic sections were studied along the

5 Atenquique ravine and its main tributaries (Fig. 2). Deposits cover a minimal area of 1.2 km², and with an average thickness of 4 m, a minimum volume of $3.2 \times 10^6$ m³ was estimated for the flow.

The main flood probably formed in the Atenquique ravine, but was enhanced by the confluence of flows from its tributaries: Dos Volcanes at 11.2 km, Arroyo Seco and Los Plátanos, at 22.5 km. During the first 10 km (as recorded by the proximal exposures) the flow moved down steep slopes, eroding and incorporating coarse alluvium and sand. Downstream, in the medial

exposures, the flow encountered gentler slopes, reducing its velocity and promoting deposition of part of the sediment load. Just upstream of the village, below the junction with the Arroyo Seco and Los Plátanos ravines, eyewitnesses reported peak flood levels, possibly enhanced by the engulfing of a small water reservoir. At the junction, the flow captured the fine-material load of flow from the Arroyo Seco and Plátanos ravines, causing significant dilution and a sudden increase in the flow turbulence. Downstream from the town, the flow lost its capacity to transport large boulders, probably due to widening of the flow and the

consequent fall in velocity, which was further reduced by the hydraulic roughness effects of the flow impacting buildings. The diluted flow probably had a velocity in the range of 4 to 6 m/s, obtained by comparison with analogous flows (Pierson, 1985; Saucedo et al., 2008). The flood finally continued downstream to join with the perennial Tuxpan River, where it emplaced up to 6 m of deposits.

## 2.2 Multiple sources and their locations

The 1955 debris flow, according to eyewitness accounts and deposit analyses, emanated from multiple sources throughout the watershed. The existence of multiple source areas presents a unique challenge when attempting to model the flow. Eyewitnesses confirm that after the event, many small landslides scars were present along the main ravine and its tributaries. It is hypothesized that these landslides, triggered by rain infiltration, supplied the bulk of the material (Saucedo, 2003; Saucedo et al., 2008). To account for this, and based on the work in Rupp (2004) we initiate the flow from five major source locations, reported in Fig. 2. We remark that our numerical simulation toolkit allows for multiple starting points in the same run. Each source consists of a paraboloid pile of material with unitary aspect ratio. Two source locations (#1 and #2) are placed in the main ravine, one (#3) in a lateral valley associated with the regional Tamazula fault, one source (#4) in Arroyo Seco and one (#5) in Arroyo Plátanos. These locations are selected based upon local topography. The first three are located on steep upland slopes, while the other two are located along the main drainage due to the lack of steep terrain nearby (Rupp, 2004). The partition of volume $V = \sum_{k=1}^{5} V_k$ among the five sources is scaled based upon the size of the drainage basin they are located within. In particular, $\forall k$ we define $w_k = V_k/V$ as:

$$w_1 = w_3 = w_4 = 19.24\%, \quad w_2 = 37.58\%, \quad w_5 = 4.70\%.$$

This is equivalent to choosing pile radii of $80\ m$, $100\ m$ and $50\ m$ respectively.





We remark that source number, location, and volume partition will be preserved in the following analysis. However, we tested whether small variations affect the character of the simulated flow only proximally. Large variations, for example additional testing focused on increasing the weight $w_4$ of the source in Arroyo Seco are not excluded, but would require additional field work to be constrained, and are beyond the purpose of this study. More details about volume $V$ are provided in Section 3.1.

## 3 Prediction-oriented probabilistic modeling

Our numerical modeling of the Atenquique flow proceeds by first assuming that the laws of mass and momentum conservation hold for properly defined system boundaries. The flow had very small depth compared to its length, and hence we assume that it is reasonable to integrate through the depth to obtain simpler and more computationally tractable equations (Savage and Hutter, 1989). The depth-averaged Saint-Venant type equations that result are:

$$
\begin{aligned}
\frac{\partial h}{\partial t} + \frac{\partial}{\partial x}(h\bar{u}) + \frac{\partial}{\partial y}(h\bar{v}) &= 0 \\
\frac{\partial}{\partial t}(h\bar{u}) + \frac{\partial}{\partial x}\left(h\bar{u}^2 + \frac{1}{2}kg_z h^2\right) + \frac{\partial}{\partial y}(h\bar{u}\bar{v}) &= S_x \\
\frac{\partial}{\partial t}(h\bar{v}) + \frac{\partial}{\partial x}(h\bar{u}\bar{v}) + \frac{\partial}{\partial y}\left(h\bar{v}^2 + \frac{1}{2}kg_z h^2\right) &= S_y
\end{aligned}
\tag{1}
$$

Here the Cartesian coordinate system is aligned such that $z$ is normal to the surface; $h$ is the flow depth in the $z$ direction; $h\bar{u}$ and $h\bar{v}$ are respectively the components of momentum in the $x$ and $y$ directions; and $k$ is the coefficient which relates the lateral stress components, $\bar{\sigma}_{xx}$ and $\bar{\sigma}_{yy}$, to the normal stress component, $\bar{\sigma}_{zz}$. Note that $\frac{1}{2}kg_z h^2$ is the contribution of hydrostatic pressure to the momentum fluxes. $S_x$ and $S_y$ are the sum local stresses: their definition depends on the constitutive model of the flowing material (Kelfoun, 2011). These include the gravitational driving forces, the basal friction force resisting to the motion of the material, and additional forces specific of rheology assumptions.

In this study we adopt the *Mohr-Coulomb* (MC), *Pouliquen-Forterre* (PF) and *Voellmy-Salm* (VS) models, detailed in Appendix A. These three models for large scale mass flows are incorporated in our large scale mass flow simulation framework TITAN2D (Patra et al., 2005). The 4[th] release of TITAN2D [2] offers multiple rheology options in the same code base. The availability of three distinct models for similar phenomena in the same tool provides us with the ability to directly compare outputs and internal variables in all the three models and control for usually difficult to quantify effects like numerical solution procedures, input ranges and computer hardware (Patra et al., 2018b).

### 3.1 Preliminary definition of the input space

The definition of the input space hierarchy $\mathbb{R}_+^{d_j} \supseteq \Omega_0^j \supseteq \Omega^j \supseteq \Omega_i^j, \forall i, j$, is a fundamental part of our approach. The dimensionality $d_j$ is a characteristic of the model. The input spaces of MC and VS have three dimensions, and are therefore parameterized:

$$
\Omega_0^{\mathrm{MC}} = \left\{ (\phi_{bed}, \phi_{int}, V) \in \mathbb{R}_+^3 \right\},
$$

---

[2]available from vhub.org





$$\Omega_0^{\mathrm{VS}} = \left\{ [\arctan(\mu), \log_{10}(\xi), V] \in \mathbb{R}_+^3 \right\}.$$

On the other hand, the input space of model PF originally has six dimensions - $(\phi_1, \phi_2, \phi_3, \beta, L, V)$. Following Pouliquen and Forterre (2002), we constrain $\phi_3 = \phi_1 + 1°$. Moreover, preliminary testing in our case study showed a very similar impact from the variation of $\beta$ and $L$. So, we were able to further reduce $d_{\mathrm{PF}} = 4$, by assuming the empirical relationship:

$$\beta = f(\phi_2) := \frac{\phi_2 - 7°}{20} + 0.1, \tag{2}$$

which is consistent with the $\beta$ values presented in literature (Pouliquen and Forterre, 2002; Forterre and Pouliquen, 2003). In the following we also show the effect of fixing the value of $\beta$. We thus effectively parameterize:

$$\Omega_0^{\mathrm{PF}} = \left\{ (\phi_1, \phi_2, L, V) \in \mathbb{R}_+^4 \right\}.$$

### 3.1.1 General input space

The input space boundaries $\left( a_{k,M_j}, b_{k,M_j} \right)_{1 \le k \le d_j}$ of $\Omega_0^j$ are constrained by the general assumptions:

- **Total Volume:** $V \in [3.5,\ 5] \times 10^6\ m^3$, i.e. $4.25 \pm 0.75 \times 10^6\ m^3$.

- **Input space constraints:**

  **MC** - $\phi_{bed} \ge 5°$,     $\phi_{int} \in [\phi_{bed}, 45°]$.

  **PF** - $\phi_1 \ge 1°$,     $\phi_2 \in [\phi_1 + 6°,\ \phi_1 + 18°]$,     $L \in [0.1,\ 0.5]\ m$.

  **VS** - $\arctan(\mu) \ge 1°$,     $\log_{10}(\xi) \le 4$.

In particular, a minimum volume of $3.2 \times 10^6\ m^3$ for the deposit left by the flow was obtained by Saucedo et al. (2008). We assume that it is reasonable to increase this volume by about 10% to 50% to represent the potential real volume in the simulated flow, given the lack of recovery of any deposit for the portion of the flow reaching the Tuxpan river. We exclude basal friction angles below $5°$ in MC, and $1°$ in PF and VS because of numerical instability and unphysical behavior. In MC, we constrain $\phi_{bed} < \phi_{int}$, and we exclude internal friction angles over $45°$ because these are not considered realistic. In VS we do not allow $\xi$ over $10^4$ because it produces unphysical results. In PF we constrain $\phi_1 + 6° < \phi_2 < \phi_1 + 18°$, extending the range of values presented in literature (Pouliquen and Forterre, 2002; Forterre and Pouliquen, 2003). The parameter $L$ is related to the particle size Forterre and Pouliquen (2003), hence we define it consistent with the observed average clasts sampled in the field (Saucedo et al., 2008).

### 3.1.2 Specialized input space

The construction of $\Omega_0^j$ relies on extensive testing of the models over the general input space defined above. We base our analysis on two qualitative properties that any realistic flow must have: (i) the flow must reach the town of Atenquique in a reasonable time, (ii) the flow does not over-top the ravine walls. We quantitatively re-formulate these properties as:





(i) the flow reaches a minimum elevation $\zeta < 1200$ m a.s.l. before $t = 1200$ s,

(ii) the maximum over-spill at the confluence of flows from different sources is $< 0.1$ m thick.

In (i), we selected 1200 m elevation because it is located about 1 km upstream from the village (Saucedo et al., 2008), and $t = 1200$ s, because we observed that flows that require more time are not able to realistically inundate the village. In (ii) we focused on flow confluences because they are where the major over-spilling issues take place in our tests. Small changes in this definition not significantly affect the following analysis.

## SPECIALIZED INPUT SPACES

**Mohr-Coulomb** (a)

| TABLE A | | $\varphi_{int}$ (deg) | | | |
|---|---|---|---|---|---|
| | | 30 | 35 | 40 | 45 |
| $\varphi_{bed}$ (deg) | 5.5 | 1170 | 1150 | 1120 | 1100 |
| | 6 | 1230 | **1200** | **1180** | **1130** |
| | 6.5 | 1260 | 1230 | **1200** | **1160** |
| | 7 | 1280 | 1250 | 1220 | |

**Voellmy-Salm** (b)

| TABLE C | | $\log_{10}(\xi)$ | | | |
|---|---|---|---|---|---|
| | | 2.5 | 3.0 | 3.5 | 4.0 |
| atan($\mu$) (deg) | 1 | 1220 | **1020** | **980** | 970 |
| | 1.5 | 1260 | **1060** | **1010** | **990** |
| | 2 | 1330 | **1200** | **1060** | **1010** |
| | 2.5 | 1390 | 1280 | **1170** | **1050** |
| | 3 | | 1360 | 1280 | **1200** |
| | 3.5 | | | 1370 | 1280 |

**Pouliquenne-Forterre (1)**   $\beta$=0.1   $\varphi$2=7 (deg)   (c)

| TABLE B1 | | L (m) | | | | |
|---|---|---|---|---|---|---|
| | | 0.1 | 0.2 | 0.3 | 0.4 | 0.5 |
| $\varphi_1$ (deg) | 1 | | 1050 | 1080 | **1110** | **1130** |
| | 1.5 | | 1090 | 1120 | **1150** | **1170** |
| | 2 | | 1140 | **1170** | **1200** | 1220 |
| | 2.5 | | 1210 | 1230 | 1250 | 1260 |
| | 3 | | | | | |

**Pouliquenne-Forterre (2)**   $\beta$=0.3   $\varphi$2=11 (deg)

| TABLE B2 | | L (m) | | | | |
|---|---|---|---|---|---|---|
| | | 0.1 | 0.2 | 0.3 | 0.4 | 0.5 |
| $\varphi_1$ (deg) | 1 | **990** | **1020** | **1070** | **1110** | **1150** |
| | 1.5 | **1020** | **1070** | **1120** | **1170** | **1200** |
| | 2 | **1070** | **1130** | **1190** | 1220 | 1250 |
| | 2.5 | **1150** | 1220 | 1260 | 1280 | |
| | 3 | 1230 | 1290 | | | |

**Pouliquenne-Forterre (3)**   $\beta$=0.5   $\varphi$2=15 (deg)

| TABLE B3 | | L (m) | | | | |
|---|---|---|---|---|---|---|
| | | 0.1 | 0.2 | 0.3 | 0.4 | 0.5 |
| $\varphi_1$ (deg) | 1 | **990** | **1020** | **1090** | **1130** | **1170** |
| | 1.5 | **1020** | **1090** | **1150** | **1200** | 1230 |
| | 2 | **1070** | **1160** | 1220 | 1260 | 1290 |
| | 2.5 | **1170** | 1250 | 1280 | | |
| | 3 | 1250 | 1330 | | | |

**Pouliquenne-Forterre (4)**   $\beta$=0.7   $\varphi$2=19 (deg)

| TABLE B4 | | L (m) | | | | |
|---|---|---|---|---|---|---|
| | | 0.1 | 0.2 | 0.3 | 0.4 | 0.5 |
| $\varphi_1$ (deg) | 1 | **990** | **1050** | **1110** | **1170** | 1220 |
| | 1.5 | **1020** | **1100** | **1170** | 1230 | 1280 |
| | 2 | **1090** | **1190** | 1250 | 1300 | |
| | 2.5 | **1190** | 1270 | 1340 | | |
| | 3 | 1260 | 1360 | | | |

Table 1: Input space exploration and testing of (a) MC, (b) VS, (c) PF models, $V = 4.18 \times 10^6$ $m^3$. The 3D input space of PF is described by four 2D subspaces. Values reported are the minimum elevation $\zeta$ reached at $t = 1200$ s. A gray background marks those inputs that generate unphysical flow run-up and over-spill the ravine walls, light gray if $> 0.1$ m, dark gray if $> 1$ m. Red lines mark the subdomain[j] of flows with $\zeta \leq 1200$ m and without significant over-spill issues.

Table 1 reports the minimum elevation $\zeta$ inundated at 1200 s, and a summary of over-spill issues. These values are generated with reference to a flow volume of $V = 4.18 \times 10^6$ $m^3$, roughly equivalent to the mean value of our range, and so the input spaces have two dimensions in MC and VS, and three in PF. Following the empirical falsification principle, we snip the input range by deleting those regions that do not satisfy our requirements. The subspaces obtained by this procedure do not have rectangular shape, but look like parallelograms - this is because the effects of the input variables on the output are not independent. Maximum flow height maps of all these tests are included in Supporting Information SI1-SI3.







**Figure 3.** Overview of the specialized experimental design in (a-b) MC, (c-d) PF, (e-f) VS models. (a-c-e) are projected along the $V$ coordinate, and (b-d-f) along $\phi_{int}$, $\phi_2$ and $\xi$ coordinates, respectively. In (c-d), the color expresses the distance along the third dimension.




- In MC we observe over-spill issues if $\phi_{bed} < 6°$, and a runout that is too short if $\phi_{bed} > 6.5°$. We also note an increase in the runout distance as $\phi_{int}$ increases. In particular, $\phi_{int} \geq 35°$ is required to inundate the village.

- In VS the over-spill is observed in the region $\{\arctan(\mu) < 1.5°\} \times \{\log_{10}(\xi) > 3.5\}$.
  In contrast, the region $\{\arctan(\mu) > 3°\} \times \{\log_{10}(\xi) < 3\}$ produces runouts that are too short.

5   - Model PF must be treated with greater care because of its higher dimensionality. We divide its behavior along four different hyperplanes, corresponding to different values of $\phi_2$ and $\beta = f(\phi_2)$. Over-spill was observed only in the hyperslice with $\phi_2 = 7°$, and $L < 0.4$ m. In general, $\phi_1 < 3°$ is required to have a long-enough runout.



**Figure 4.** Overview of modified specialized experimental designs in PF, supported over the four different hyperplanes described in Table 1. All plots are projected along the $V$ coordinate.





## 3.2 Probability measures and specialized experimental design

Initially, $\forall j$ we uniformly distribute the measure $P_0^j$ supported in the general input space $(\Omega_0^j, \mathcal{F}_0^j)$:

$$P_0^j\left(p_1^j, \ldots, p_{d_j}^j\right) \sim \bigotimes_{k=1}^{d_j} Unif(a_{k,M_j}, b_{k,M_j}), \qquad (3)$$

where $(p_1^j, \ldots, p_{d_j}^j)$ is the parametrization of model $M_j \in \mathcal{M}$ described above, and $\forall k$, $(a_{k,M_j}, b_{k,M_j})$ is the input range. Latin

Hypercube Sampling is performed over $[0,1]^{d_j}$.

We enhance the sampling procedure by relying on orthogonal arrays (Owen, 1992a; Tang, 1993; Patra et al., 2018b). Those dimensionless samples are thus linearly mapped over the required intervals, providing the general experimental design. Then, according to the definition of the specialized input space, we remove the design points which lie outside of the boundary. The new design is distributed according to the subspace probability measure $P^j$, naturally defined over the $\sigma$-field $\mathcal{F}^j :=$

$\left\{B \cap \Omega^j : B \in \mathcal{F}_0^j\right\}$.

Figure 3 displays the plot of this specialized experimental design. In PF, the design is embedded in $\mathbb{R}^4$, and in Figure 4 we show the plot of ancillary experimental designs supported over the four hyperplanes described in Table 1, corresponding to different values of $\phi_2$ and $\beta = f(\phi_2)$. In the following, our time domain is $T = [0, 2400\ s]$, which provides sufficient time for simulated flows to realistically inundate the village, considering the inputs in $\Omega^j$. We call $t_f = 2400$ s, the ending time of the

simulation.

Figure 5 summarizes all the steps of our approach, following the notation defined in the Introduction. We remark that the comparison of observed data to the statistical summary of the outputs can provide some information on possibly untested plausible flows, and on the hypothetical necessity of implementing additional models.

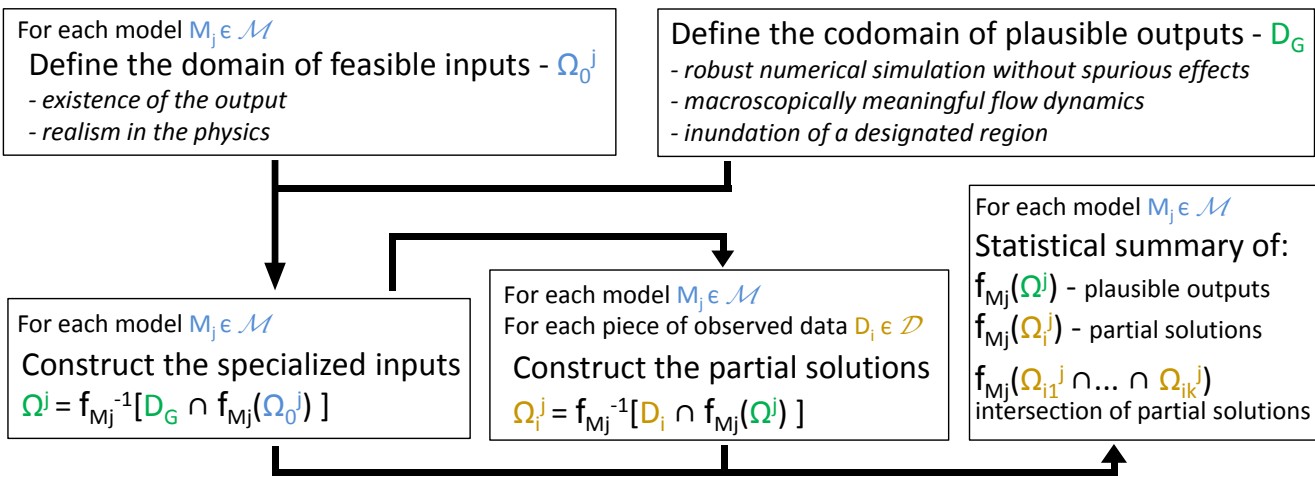

**Figure 5.** Diagram of the steps of our meta-modeling approach. Our statistical summary includes the local analysis if contributing variables.





## 4 Observable outputs and contributing variables

We devise multiple statistical measures for analyzing the data according to the specialized LHS design described in the previous section. In general, for each $M_j \in \mathcal{M}$, we sample the model inputs in a Monte Carlo simulation, and the output of each sample run is calculated as a function $f_j(\omega, \underline{\mathbf{x}}, t)$, where $\omega$ is the input, $t$ is the time and $\underline{\mathbf{x}}$ is a spatial element of the computational

grid. The family of these functions naturally defines a random variable which expresses the model outputs with respect to the probability distribution $P^j$ over the input space $(\Omega^j, \mathcal{F}^j)$. This analysis generates a tremendous volume of data that we analyze using statistical methods. The results are summarized by a family of spatial maps and temporal graphs, displaying the expectation of the model outputs and also their $5^{\text{th}}$ and $95^{\text{th}}$ percentiles, with respect to $P^j$.

In particular, we select five sites and we gather detailed results from our simulations in those special locations. These are

10 called Sites #1-#5 (stars in Figure 1). They all belong to the set of the sixty sections studied in Saucedo et al. (2008), and the corresponding section numbers are also reported. In detail they are:

 – **Site #1**, section 23, UTM 656690.1N, 2160985.4E. Along the main ravine ∼6 km upstream from the Atenquique village.

 – **Site #2**, section 28, UTM 660380.8N, 2162530.8E. In Arroyo Seco tributary ∼2.5 km upstream from the confluence.

These first two are not described in detail, but the related results are included in Supporting Information SI4-SI5. Conversely,

we focus our analysis on the other three points, all placed along the main ravine in proximity to Atenquique village.

 – **Site #3**, section 21, UTM 660258.1N, 2161315.2E. It is ∼2 km upstream from the village;

 – **Site #4**, section 17, UTM 662453.1N, 2160360.1E. Immediately upstream from the village, close to the confluence with Arroyo Seco and Arroyo Plátanos tributaries;

 – **Site #5**, section 42, UTM 663539.0N, 2160200.0E. In the village, ∼1 km downstream from the previous site.

We note that close to Site #4 there are the supports of the new bridge of the freeway to the city of Colima.

### 4.1 Percentile maps of maximum flow depth and kinetic energy

First of all, we report the spatial maps of maximum flow depth, $h$, and kinetic energy, $\kappa$:

$$H := \max_{t \in T} h(x, y), \quad K := \max_{t \in T} \kappa(x, y). \tag{4}$$

In our depth-averaged approach the *kinetic energy* [3] is defined as:

$$\kappa := \frac{1}{2} \frac{(h\bar{u})^2 + (h\bar{v})^2}{h}, \tag{5}$$

where $h\bar{u}$ and $h\bar{v}$ are, respectively, the components of momentum in the $x$ and $y$ directions.

---

[3]while the use of dynamic pressure is more common in these applications, we have used kinetic energy as a more stable indicator of potential to damage infrastructure and a quantity whose computation as a conserved scalar across all models is more stable.



This is equivalent to $\frac{1}{2}h||(\bar{u},\bar{v})||^2$, that is half the flow height times the speed square. Thus $\kappa$ is formally the kinetic energy density per unit of surface area, for a mass with unit density.

## MAXIMUM FLOW HEIGHT PERCENTILE MAPS

**Figure 6.** Maximum flow height $H$ as a function of time in (a-b) MC, (c-d) PF, (e-f) VS model. (a-c-e) are the $5^{th}$ and (b-d-f) are the $95^{th}$ percentile values with respect to $P^j$. Colors are related to the flow height. Elevation contours are included at intervals of 100 m (gray) and 500 m (black) (NASA (JPL), 2014). Sites #3,#4,#5 are displayed.

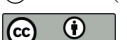



## MAXIMUM KINETIC ENERGY PERCENTILE MAPS

**Figure 7.** Maximum kinetic energy $K$ as a function of time in (a-b) MC, (c-d) PF, (e-f) VS model. (a-c-e) are the $5^{th}$ and (b-d-f) are the $95^{th}$ percentile values with respect to $P^j$. Colors are related to the energy in logarithmic scale. Elevation contours are included at intervals of 100 m (gray) and 500 m (black) (NASA (JPL), 2014). Sites #3,#4,#5 are displayed.

The kinetic energy, $K$, in a traditional sense may be calculated over an arbitrary region $R$ as:

$$\text{Kin}(R) = \rho \int_R \kappa(x,y)\,dx\,dy = \rho \int_R \frac{1}{2} h(x,y) \left[ \bar{u}(x,y)^2 + \bar{v}(x,y)^2 \right] dx\,dy,$$





where $\rho$ is the density of the flow, typically in $[1000, 2000]$ $kg/m^3$, depending on the flow water content.

Figure 6 reports the maximum flow depth, $H$, and Figure 7 the maximum kinetic energy, $K$. MC shows the lowest values of both flow depth and energy, while VS the highest, especially in the distal part of the domain. In MC, the flow in the tributaries is not capable of reaching the village, while in the $95^{th}$ percentile maps of PF and VS, it is. In VS, the flow in Arroyo Plátanos joins the main ravine even in the $5^{th}$ percentile map. In general, local maxima of flow depth are located in the ravine, while the kinetic energy shows a more regular decrease. The energy values at the head of the flow are in $[1, 10]$ $m^3/s^2$, meaning a relatively slow flow compared to the dynamics observed upstream. Significant over-spill issues are absent, but in the $95^{th}$ percentile maps all the models report some flow in the Dos Volcanes ravine, of about $[1, 5]$ $m$ maximum height, and slightly less in PF.

## 4.2 Local properties of the flow

We further analyze the flow properties at the three sites selected above, and located in the distal part of the flow (Sites #3,#4, #5), with very different results. All the quantities reported are estimated on the element of the grid which contains the coordinates of the site. We remark that the grid is adaptive, and hence the values can be affected by the size and position of the element. However, the integration over the input space significantly reduces this effect (Patra et al., 2018b).

Along with the locally analyzed flow height and speed, we calculate the local *contributing variables* in the modeling equations, that is, the *dominance factors* and the *expected contributions* related to the force terms in the conservation laws that characterize the models. In particular, $\forall n = 1, \dots, N$ the dominance factor, $p_n$, is the probability that the related force term, $F_n$, is largest. The no-flow probability, $p_0$, is also included, and $\sum_{n=0}^{N} p_n = 1$. In contrast, $\forall n$ the expected contribution $E^{P_j}[C_n]$ is the mean of the force term, $F_n$, divided by the greatest (dominant) term. It belongs to $[0,1]$ and represents the degree of relevance of the $n^{th}$ term with respect to the dominant one. Contributing variables and their statistics are introduced in Patra et al. (2018b) and detailed in Appendix B.

The higher dimensionality of $\Omega^{\mathrm{PF}}$ requires additional testing. In particular, flow height and speed are estimated on the hyperplane $\beta = 0.5$, $\phi_2 = 15°$, and the probability distributions are not remarkably different. Additional plots concerning sites #1 and #2 are included in SI4-SI5.

Figure 8 shows the local properties at site #3, located $\sim$2 km upstream from Atenquique village. In MC, the flow reaches the site in $[600, 900]$ s, with a flow height of initially $[5, 7]$ m, and $[3, 5]$ m at $t_f = 2400$ s. Flow speed is initially in $[1.5, 4.5]$ $m/s$, and one-third of these values are at $t_f$. Only the basal friction can be dominant, with gravitational force being 40% of it, and internal friction 25%, on average. In PF the flow reaches the site at $[300, 1000]$ s, with an initial flow height of $[4, 8]$ m, having a short-lasting peak of 10 m in the $95^{th}$ percentile, and $[3, 5]$ m at $t_f$. Flow speed is initially in $[4, 6]$ $m/s$, and has a short peak of almost $14$ $m/s$ in the $95^{th}$ percentile. It becomes $[2.0, 2.5]$ $m/s$ at $t_f$. Either basal friction, gravity, or pressure force (related to the gradient of thickness) can be dominant. Gravitational force adds 80% of the expected contribution, while basal friction 70% and pressure force 45%. In VS, the flow first reaches the site in $[300, 800]$ s, with a flow height of $[1.5, 12]$ m, initially, and a short peak at 14 m, becoming $[1.5, 5]$ at $t_f$. Flow speed is in $[1, 15]$ $m/s$, initially, has a short peak at 20 $m/s$, and then becomes $[1.5, 3.5]$ $m/s$ at $t_f$. Either basal friction or gravity can be dominant, both with 75% expected contribution.





We note that gravity tends to slightly decrease in expected contribution, as well as its dominance factor, through time. This is a numerical effect related to the variation of the mesh, which changes our approximation of local slope.

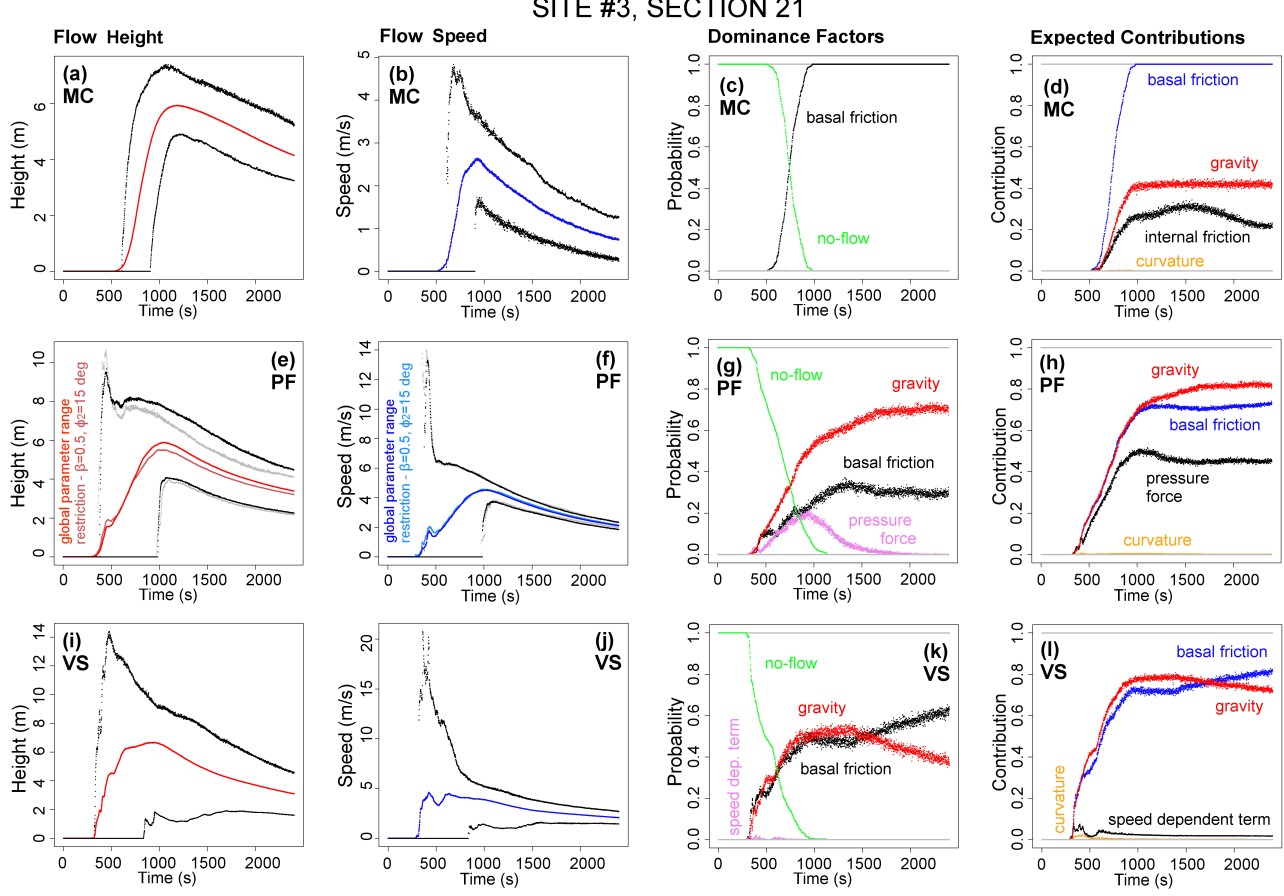

**Figure 8.** Local flow properties in Site #3, ∼2 km upstream from Atenquique village. (a,e,i) show flow height, (b,f,j) flow speed, (c,g,k) dominance factors, (d,h,l) expected contributions of the force terms. Different models are plotted separately: (a-d) assume MC; (e-h) assume PF, and (e,f) include estimates on a hyperplanar restriction of the input domain; (i-l) assume VS. In (a,b,e,f,i,j) colored line is mean value, black/gray lines are $5^{th}$ and $95^{th}$ percentile bounds.

In summary, MC is characterized by a lower speed, and by the dominance of basal friction. The expected contribution of the internal friction is not negligible, meaning the internal shear of the material is important. In PF, the pressure force contribution is significant, and it can even be the dominant force initially. This is related to the steepening of the flow front. An initial short lasting wave of high speed is observed in either PF or VS, as is particularly evident in the upper bound of the plots. This fast wave is related to the closest source, #3. The uncertainty affecting height and speed is generally higher in VS than in PF, in spite of the higher dimensionality of the second.



Figure 9 shows the local properties at site #4, located immediately upstream from Atenquique village, and close to the supports of the new bridge of the freeway to the city of Colima. In MC, the flow reaches the site in [1300, 2100] s, with a flow height of [2, 5.5] m. Flow speed is $< 0.6\ m/s$, and $0.1\ m/s$, on average. Only basal friction is dominant, with the gravitational force being 30% of basal friction, and internal friction 10%, on average. In PF, the flow reaches the site at [600, 1800] s, with a flow height of [3, 6] m, which is stable in time. Flow speed is initially in [1, 4] $m/s$ and half of these values are at $t_f$. Either basal friction or gravity can be dominant. Gravitational force can be 95% of the expected contribution, while basal friction is 65%, and the pressure force, 25%.

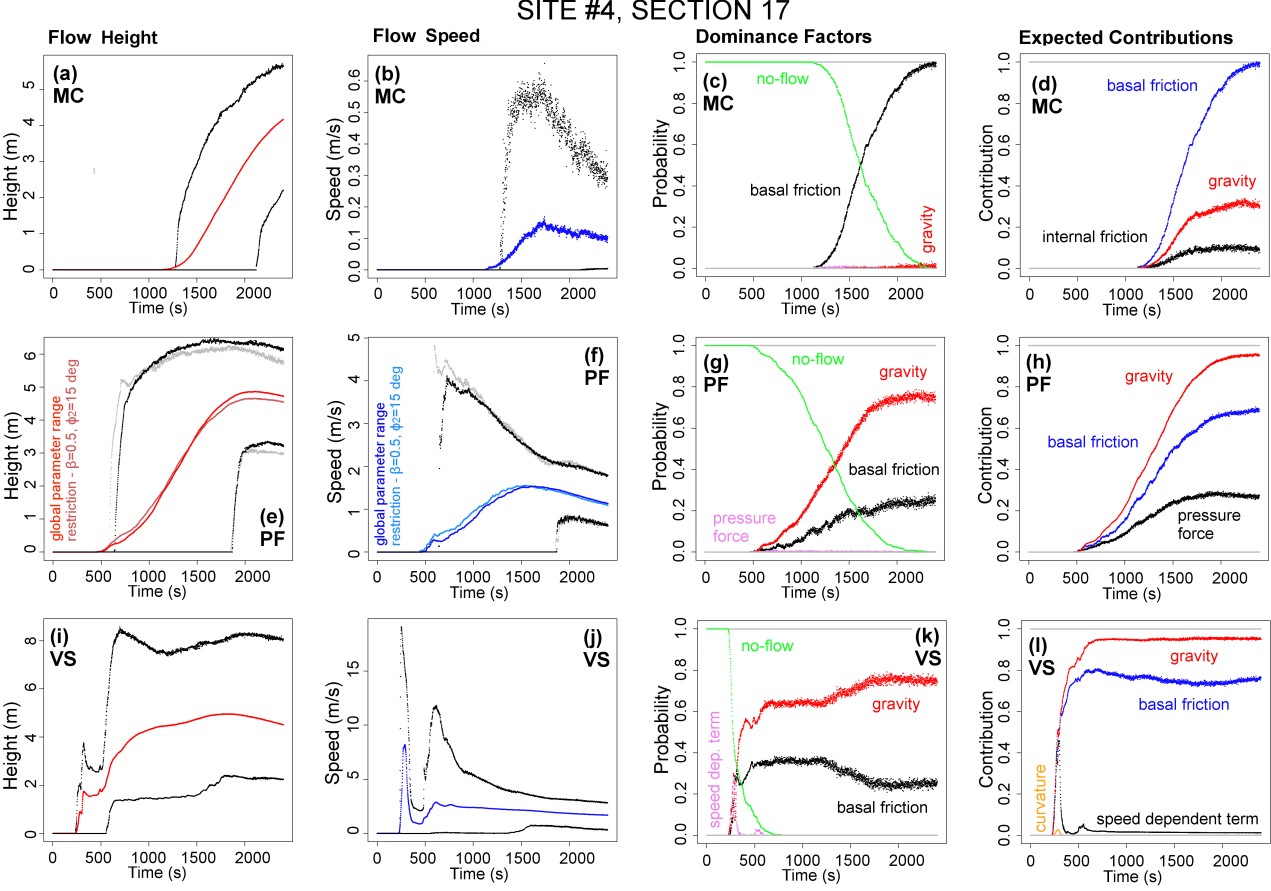

**Figure 9.** Local flow properties in Site #4, immediately before Atenquique village. (a,e,i) show flow height, (b,f,j) flow speed, (c,g,k) dominance factors, (d,h,l) expected contributions of the force terms. Different models are plotted separately: (a-d) assume MC; (e-h) assume PF, and (e,f) include estimates on a hyperplanar restriction of the input domain; (i-l) assume VS. In (a,b,e,f,i,j) colored line is mean value, black/gray lines are $5^{th}$ and $95^{th}$ percentile bounds.

In VS, the flow reaches the site in [250, 550] s, with a first wave of $< 3$ m. A second wave of [1.5, 8] m arrives, and stays stable in depth. Flow speed also shows two waves - it is $< 19\ m/s$, initially, then rises to $< 12\ m/s$, with a gap in the





middle. It is $[1, 3]$ $m/s$ at $t_f$. Either gravity or basal friction can be dominant, with 95% and 80% of the expected contribution, respectively. In the initial fast wave, the speed dependent friction can be dominant, with 25% of the expected contribution.

In summary, all the models show lower flow height and speed than at the previous site upstream. The flow depth is stable, without decreasing downstream, meaning no formation of a significant deposit. MC is much slower than the other models, and
5  its dynamics is completely dominated by basal friction. An initial, short-lasting wave of high speed is observed in VS. This fast wave is related to source #5 in Arroyo Platános.

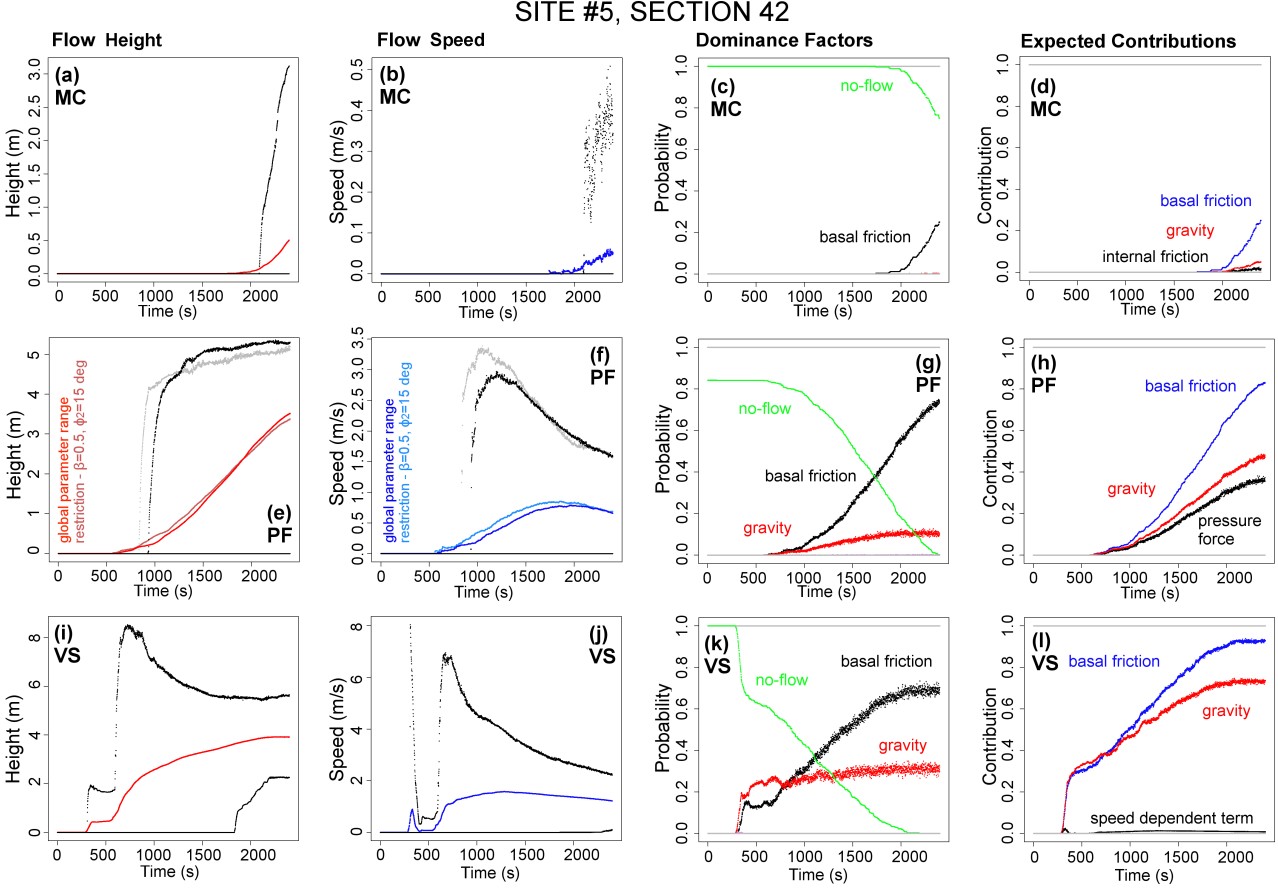

**Figure 10.** Local flow properties in Site #5, ∼1 km downstream inside Atenquique village. (a,e,i) show flow height, (b,f,j) flow speed, (c,g,k) dominance factors, (d,h,l) expected contributions of the force terms. Different models are plotted separately: (a-d) assume MC; (e-h) assume PF, and (e,f) include estimates on a hyperplanar restriction of the input domain; (i-l) assume VS. In (a,b,e,f,i,j) colored line is mean value, black/gray lines are $5^{\mathrm{th}}$ and $95^{\mathrm{th}}$ percentile bounds.

Figure 10 shows the local properties at site #5, ∼1 km downstream from site #4, inside the bounds of Atenquique village. In MC, the flow only starts to inundate the site after 2100 s, with a flow height $< 3$ m. Flow speed is $< 0.5$ $m/s$, and ten times lower on average. Only basal friction can be dominant. In PF, the flow reaches the site at $[900, 2400]$ s, with a flow height



< 5 m. Flow speed is initially < 3 $m/s$, and less than half of this value at $t_f$. Basal friction dominates the dynamics, with 45% expected contribution from the gravitational force, and 35% from the pressure force. In VS, the flow reaches the site in [300, 600] s, with a first wave of < 2 m, and then a second wave < 8 m. This second wave becomes [2, 5.5] m at $t_f$. Flow speed is < 8 $m/s$ in the first wave, and then < 7 $m/s$ in the second, with a gap in the middle. The average speed is stable at

1.5 $m/s$. Either basal friction or gravity can dominate, with 95% and 70% expected contributions, respectively.

  In summary, the models show a decrease in flow height and speed compared to the previous site. In MC, the site is not always reached, and in PF some input values inundate the site only at the end of the time domain. In PF, restriction of the inputs over a lower dimensional subspace produces a lag of about 100 s, possibly motivated by a fixed value of $\phi_2 = 15°$ being higher than the middle of its variable range. The fast wave in VS is again related to source #5.

## 5   The likelihood of a model given uncertain data

Figures 11 and 12 show the flow height and speed histograms at the three selected sites, either their maximum value, or their value at $t = 2400$ s. These confirm what is summarized in the previous section. The following probability density values are evaluated with respect to m and m/s, respectively. In Figure 11:

- **Site #3** Flow height pdf at $t_f$ is above 0.1 over: [3, 5.5] m in MC, [2, 4.5] m in PF and [1.5, 4.5] m in VS. The first two
models are more peaked, while the third produces a tail of very large thickness values, up to 8 m. The maximum height
    pdf is above 0.1 over: [5.5, 8] m in MC, [6, 9] m in PF, [8, 13] in VS.

- **Site #4** Flow height pdf at $t_f$ is above 0.1 over: [2.5, 6] m in MC, [3, 6] m in PF and [2, 7] m in VS. The maximum
    height pdf is above 0.1 over: [2.25, 6] m in MC, [5, 7] m in PF, [6, 9.5] m in VS. The three models show well separate
    modal values at 4, 6 and 8 m, respectively.

- **Site #5** In MC and PF do not always reach the site in $T = [0, 2400]$ s, so they both show a modal value of flow height in
    0 m. The flow height pdf at $t_f$ is above 0.1 over: [1, 1.5] m in MC with a tail up to 3.5 m, [3, 5.5] m in PF, [2, 6] m in
    VS. The maximum height pdf is almost equal to final height in MC, and it is above 0.1 over: [3.5, 6] m in PF, [3.5, 9] m
    in VS. Both PF and VS show a modal value at 5 m.

In Figure 12:

- **Site #3** Flow speed pdf at $t_f$ is above 0.25 over: [0.1, 1.4] m/s in MC, [1.6, 2.4] m/s in PF and [1.4, 2.8] m/s in VS. MC
    is bimodal at 0.5 m/s and 1 m/s, while PF has a modal value at 2.1 m/s. The maximum speed pdf is above 0.05 over:
    [1, 7] m/s in MC, [4, 8] m in PF, [4, 16] in VS. PF and VS have tails up to 25–30 m/s.

- **Site #4** Flow speed pdf at $t_f$ is above 0.25 over: [0.00, 0.45] m/s in MC, [0.6, 1.8] m/s in PF and [0.5, 3.0] m/s in VS. PF
    has a modal value at 1 m/s. Maximum speed pdf in MC is concentrated below 1 m/s. In PF, it is above 0.25 over [1, 5]
30    m/s. In VS, it is bimodal and above 0.25 over: [2, 6] m/s and [12, 20] m/s.





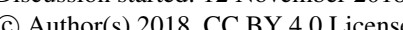

**Figure 11.** Histograms of local flow height in Sites #3,#4, #5. (a,c,e) show height at $t = 2400$ s, (b,d,f) maximum height. Different models are displayed with different colors. Dots on the height axis show the uncertainty interval of data - green is the deposit thickness of Saucedo et al. (2008) and unpublished data, violet is the wave height documented by the survivors.

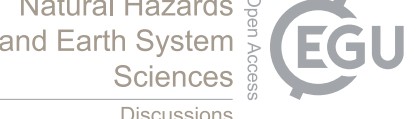

**Figure 12.** Histograms of local flow speed in Sites #3,#4, #5. (a,c,e) show speed at $t = 2400$ s, (b,d,f) maximum speed. Different models are displayed with different colors. Cyan dots on the speed axis show the uncertainty interval of estimated speed Pierson (1985).



– **Site #5** Flow speed pdf at $t_f$ is above $0.25$ over: $[0.0,\ 0.4]$ m/s in MC, $[0.0,\ 1.4]$ m/s in PF and over $[0.0,\ 0.2]$ and $[0.4,\ 2.2]$ m/s in VS. The maximum speed pdf is above $0.05$ over: $[0.0,\ 0.2]$ and $[0.4,\ 1.2]$ m/s in MC, to $[0,\ 4]$ m in PF, $[0,\ 9]$ in VS. PF and VS have tails up to $8$ and $13$ m/s, respectively.

## 5.1 Alternative performance scores of the models

5   In Figures 11 and 12, we display the empirical data concerning the observed quantities. These intervals are examples of uncertain data to test our methodology. Further research could refine/modify them.

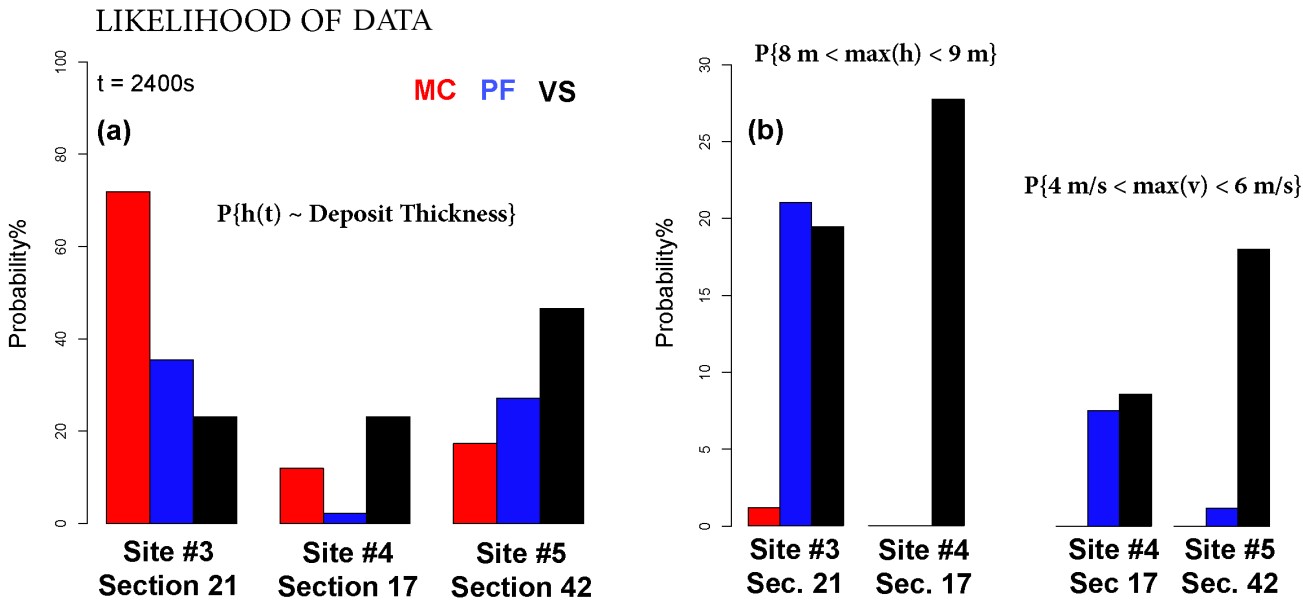

**Figure 13.** Barplots of data likelihood in Sites #3, #4, #5. (a) compares flow height at $t = 2400$ s with observed deposit thickness (Saucedo et al., 2008). (b) compares maximum height and maximum speed with observed wave height (Ponce-Segura, 1983) and analog flow speed (Pierson, 1985). Different models are displayed with different colors.

1. The deposit thickness, calculated from the envelope of the closest field sections, is
   $[3.7,\ 5.5]$ m at **Site #3**, $[1.7,\ 3]$ m at **Site #4**, and $[1.4,\ 3.8]$ m at **Site #5** (Saucedo et al., 2008).

2. the flow height in Atenquique village, from historical documents and witnesses, is
10   $[8,\ 9]$ m at **Site #3** and/or **Site #4** (Ponce-Segura, 1983; Saucedo et al., 2008).

3. The flow speed following the inundation of the village, based on a comparison with analog flows, is
   $[4,\ 6]$ $m/s$ at **Site #4** and/or **Site #5** (Pierson, 1985; Saucedo et al., 2008).




The estimation of the likelihood of the pieces of observed data is an essential step towards the definition of partial solutions of the inverse problem. Besides this, it is also relevant information in the model selection problem. The likelihood of a data piece, $D_i$, attaining its value given a certain model is defined as $P^j(D_i), \forall i \in I, \forall M_j \in \mathcal{M}$. We remark that this is not a pdf value, but the probability of a measurable set. Figure 13 shows the barplots of data likelihood. Figure 13a considers deposit

thickness values, under the assumption that they are equivalent to the flow height at $t_f$. At Site #3, 70% of MC inputs provide flow thickness consistent with the deposit range, against 35% of PF and 20% of VS inputs. At Site #4, likelihood scores are lower: 20% in MC, 10% in VS, <5% in PF. At Site #5, instead, we have: 15% in MC, 25% in PF, 45% in VS. Figure 13b evaluates the maximum flow height when the flow entered Atenquique village. In Site #3, 20% of inputs in PF and VS provide results consistent with data, while only <5% do so in MC. At Site #4, the likelihood score of VS is 27%, while it is null in the

other models. Figure 13c focuses on the maximum flow speed after the flow inundated the village. At Site #4, 8% of inputs in PF and 9% in VS provide speeds in the most likely range. At Site #5, these are only 1% in PF, and 17% in VS.

In summary, model performance is dependent on the selected quantity of interest, and on the spatial location. Regarding the deposits, MC performs well at Site #3, while VS does so at Site #5. In the evaluation of the maximum flow depth in the village, both PF and VS can replicate the values at Site #3, and only VS can replicate the values at Site #4. If we focus on the maximum

flow speed, at Site #4 both PF and VS perform moderately well, while at Site #5, only VS can provide speed values inside the assumed range.

## 6   Partial solutions in the input space

Figures 14, 15, 16 display three examples of partial solutions in the specialized experimental design. For each example $n = 1, 2, 3$ we select a subfamily of empirical data $(D_i)_{i \in I_n}$ and define, $\forall j$:

$$\tilde{\Theta}_n^j := \bigcap_{i \in I_n} \Omega_i^j. \tag{6}$$

Example #1 focuses on the deposit thickness data at Sites #3, #4, #5. Examples #2 and #3 consider the deposit thickness at Site #5 only. Additionally, Example #2 evaluates the maximum flow height at Site #4, and the maximum flow speed at Site #5, while Example #3 does that at Site #3 and Site #4, respectively.

Figure 14 concerns Example #1. In all the models, $\tilde{\Theta}_1^j = \emptyset$. In MC, the set of inputs that replicate the deposit thickness at Site #4 are disjoint from those in Site #5. Instead, in PF and VS the inputs related to Site #3 are disjoint from those related to Site #4. Figure 15 concerns Example #2. In MC and PF, the maximum flow height from the data is never reproduced. In MC, the required maximum flow speed is never achieved. Thus, $\tilde{\Theta}_2^j \neq \emptyset$ only if $j = \text{VS}$. The partial solution inputs in $\tilde{\Theta}_2^{\text{VS}}$ are bounded by:

$$\arctan(\mu) \in [1.0, \, 1.8], \quad \xi \in [3.1, 3.7], \quad V \in [3.8, \, 5.0] \times 10^6 \ m^3.$$



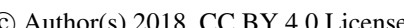

**Figure 14.** Example #1 of partial solution inputs in (a-b) MC, (c-d) PF, (e-f) VS experimental design. (a-c-e) are projected along the $V$ coordinate, and (b-d-f) along $\phi_{int}$, $\phi_2$ and $\xi$ coordinates, respectively. The color expresses the considered data: yellow is deposit thickness in Site #5, blue in Site #4, red in Site #3 (Saucedo et al., 2008).



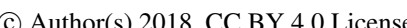



**Figure 15.** Example #2 of partial solution inputs in (a-b) MC, (c-d) PF, (e-f) VS experimental design. (a-c-e) are projected along the $V$ coordinate, and (b-d-f) along $\phi_{int}$, $\phi_2$ and $\xi$ coordinates, respectively. The color expresses the considered data: yellow is deposit thickness in Site #5 (Saucedo et al., 2008), blue is wave height in Site #4 (Ponce-Segura, 1983), red is flow speed in Site #5 (Pierson, 1985).





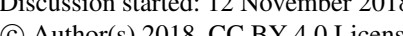

Figure 16. Example #3 of partial solution inputs in (a-b) MC, (c-d) PF, (e-f) VS experimental design. (a-c-e) are projected along the $V$ coordinate, and (b-d-f) along $\phi_{int}$, $\phi_2$ and $\xi$ coordinates, respectively. The color expresses the considered data: yellow is deposit thickness in Site #5 (Saucedo et al., 2008), blue is wave height in Site #3 (Ponce-Segura, 1983), red is flow speed in Site #4 (Pierson, 1985).




Figure 16 is related to Example #3. In MC, the required maximum flow speed from the data is never reproduced. We have that $\tilde{\Theta}_2^j \neq \emptyset$ for $j \in \{\mathrm{PF}, \mathrm{VS}\}$. In PF, the partial solution inputs in $\tilde{\Theta}_3^{\mathrm{PF}}$ are bounded by:

$$\phi_1 \in [1.0, \ 1.6], \quad L \in [0.12, \ 0.25] \ m, \quad V \in [3.9, \ 4.9] \times 10^6 \ m^3.$$

In VS, only one point belongs to the partial solution input set, with $\arctan(\mu) \simeq 1.2$, $\xi \simeq 3.1$, $V \simeq 3.6 \times 10^6 \ m^3$. Remarkably, the input spaces reproducing the three required pieces of empirical data are almost disjoint in pairs, and $\tilde{\Theta}_3^{\mathrm{VS}}$ is small and close to the frontiers of the uncertainty ranges. Additional details in PF over the hyperplane $\{\beta = 0.5, \ \phi_2 = 15\}$ described in Table 1 are included in Supporting Information SI6.

## 6.1 Examples of conditional results

The solution of the partial inverse problems can enable us to select a model, which nevertheless depends on the required properties and the spatial location. In particular:

- in Example #1 the inverse problem is not well posed,

- in Example #2 only in VS can we find solutions: $\tilde{\Theta}_2^{\mathrm{VS}} \neq \emptyset$,

- in Example #3 both in PF and VS we find solutions: $\tilde{\Theta}_3^{\mathrm{PF}} \neq \emptyset$, $\tilde{\Theta}_3^{\mathrm{VS}} \neq \emptyset$.

The points in the experimental design that belong to $\tilde{\Theta}_2^{\mathrm{VS}}$ and $\tilde{\Theta}_3^{\mathrm{PF}}$ are 21 and 9 respectively. We do not detail $\tilde{\Theta}_3^{\mathrm{VS}}$ because it contains only one design point and the results can be disrupted even by relatively small variations of the inputs. Additional tests at a finer resolution in the experimental design could be performed to achieve a more accurate characterization of the conditional input spaces, if required.

In Figures 17 and 18, we report the histograms of:

$$K = \max_{t \in T} \kappa, \quad \kappa_f := \kappa(t_f) \quad Q := \max_{t \in T} \frac{\kappa}{h}$$

at Sites #4 and #5. The spatial maps of the maximum in flow height, $h$, and kinetic energy, $\kappa$, are also displayed. The *dynamic pressure $Q$* and the *kinetic energy $\kappa$* formally assume a mass with unit density.

In Figure 17, at Site #4:

$$K \in [100, \ 400] \ m^3/s^2, \quad \kappa_f \in [4, \ 16] \ m^3/s^2, \quad Q \in [0, \ 150] \ m^2/s^2$$

and at Site #5:

$$K \in [40, \ 140] \ m^3/s^2, \quad \kappa_f \in [0, \ 9] \ m^3/s^2, \quad Q \in [7.5, \ 17.5] \ m^2/s^2$$

The modal values of $K$ are $\sim 225 \ m^3/s^2$ and $60 \ m^3/s^2$, respectively.

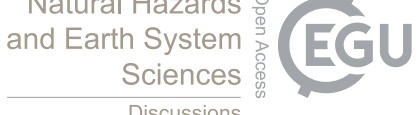



**Figure 17.** Flow properties of VS model, over the input space $\tilde{\Theta}_2^{\mathrm{VS}}$. Histograms of (a,b) local kinetic energy and (c) dynamic pressure in Sites #4 and #5, (a) at $t = 2400$ s, (b,c) maximum value. Different sites are displayed with different colors. Mean values over $\tilde{\Theta}_2^{\mathrm{VS}}$ of the maps of maximum (d) kinetic energy and (e) flow height as a function of time. Colors are related to their values. Elevation contours are included at intervals of 100 m (gray) and 500 m (black) (NASA (JPL), 2014). Sites #4 and #5 are displayed.





**Figure 18.** Flow properties of PF model, over the input space $\tilde{\Theta}_3^{\mathrm{PF}}$. Histograms of (a,b) local kinetic energy and (c) dynamic pressure in Sites #4 and #5, (a) at $t = 2400$ s, (b,c) maximum value. Different sites are displayed with different colors. Mean values over $\tilde{\Theta}_3^{\mathrm{PF}}$ of the maps of maximum (d) kinetic energy and (e) flow height as a function of time. Colors are related to their values. Elevation contours are included at intervals of 100 m (gray) and 500 m (black) (NASA (JPL), 2014). Sites #4 and #5 are displayed.





In Figure 18, at Site #4:

$$K \in [35,\ 75]\ m^3/s^2, \quad \kappa_f \in [3,\ 8]\ m^3/s^2, \quad Q \in [8,\ 13]\ m^2/s^2$$

and at Site #5:

$$K \in [18,\ 34]\ m^3/s^2, \quad \kappa_f \in [1,\ 7]\ m^3/s^2, \quad Q \in [4,\ 7]\ m^2/s^2$$

Modal values are not well-constrained in this case.

In the spatial maps, PF shows slightly lower maximum flow height, and significantly lower energy than VS, especially in the distal part of the domain. The flow in the tributaries can reach the village, except for the smallest flows of Arroyo Plátanos in PF, which however at $t_f$ are only tens of meters from the main branch. As in the unconditional maps, local maxima of flow height are located in the ravine, while the kinetic energy shows a more regular decrease.

In summary, the statistical analysis of the partial solutions told us:

– the deposit thickness is the three selected sites is not reproduced by any of the models. In particular, the input choices that fit the Site #3 are inconsistent with those that fit the deposit downstream. This advocates the possibility of testing additional models, for example including an entrainment term in the mass conservation equation.

– MC model is not capable of reproducing the required maximum flow height and speed in the village. Its feasible input space does not allow us to reduce the friction further. Even if PF can reproduce the required height and speed when impacting the village, only VS is capable of maintain those values also in the downstream part of the village.

In particular, models using MC based rheologies are unlikely to reproduce the properties of the 1955 flow. Instead, the flexible basal friction angle in PF allows for both higher speed and longer runout, consistent with those observed. The higher dimensionality of its parameter space is not significantly increasing the uncertainty affecting the outputs. Similarly, the velocity dependent term in VS is a very robust mechanism for preserving numerical stability, avoiding the spurious results that affect the MC model at equivalently low values of basal friction. Indeed the most high levels of simulated speed are observed with VS.

We remark that the assumed $[4,\ 6]\ m/s$ constraint on the maximum flow speed has an immediate effect on the dynamic pressure estimates. Imposing it at Site #5 as in the Example #2, or Site #4 as in the Example #3, can radically change the results, even if the required deposit thickness at Site #5 is not modified. Additional information on the speed properties in the village could thus allow us to further discriminate the performance of the models.

## 7 Conclusions

In this study, we have introduced a new prediction-oriented method for hazard assessment of volcaniclastic debris flows (lahars), based on multiple geophysical mass flow models. Similar strategies have been applied in hurricane hazard analysis (Krishnamurti et al., 2016; Ghosh and Krishnamurti, 2018). In particular, our approach decomposes the original inverse problem into a hierarchy of simpler problems, and allows for the exploration of the impact of synthetic flows that are similar to those that occurred in the past, but different in plausible ways.




We applied our procedure to a case study of the 1955 Atenquique volcaniclastic debris flow. We adopted and compared three depth averaged models based on the Saint-Venant equations that are widely used in hazard assessment. Namely, Mohr-Coulomb (MC), Pouliquen-Forterre (PF), and Voellmy-Salm (VS). In summary:

- We defined a *specialized experimental design* after assuming: the realism of the underlying physics, the numerical
simulation is robust in some sense, and the flow dynamics or inundation output is meaningful. This produced a range of
         output simulations that contain valuable information for hazard assessment.

   Indeed, these outputs do not strictly reconstruct past flows, hence can provide hazard estimates under constraints weaker
   than those used therein, potentially including cases of extreme events. Moreover, our designs were not trivial geomet-
   rically, due to the correlated effects of model inputs. This is a first step towards the development of an objective and
partially automated experimental design.

- We described the statistics of the outputs and contributing variables by performing a Monte Carlo simulation over the
   specialized design. We made global maps of the flows, and investigated detailed characteristics. This allowed us to
   calculate the likelihood that different model realizations reasonably represented the 1955 Atenquique flow, given mul-
   tiple pieces of field data regarding its characteristics. Depending on how it is looked at, the exercise provided useful
information in either model selection or data inversion.

   Our analysis concerned the mean values and uncertainty percentiles of quantities of interest. Moreover, the probabilistic
   setting allowed us to make inferences regarding the uncertainty affecting the data. We analyzed the contributing variables,
   which shed light on the different assumptions underlying the three considered models. In particular, the MC model is
   generally characterized by a lower speed over its feasible input space, when compared to the other models. The expected
contribution of the internal friction is significant, meaning the internal shear of the material is important. In the PF model,
   the pressure force contribution related to the steepening of the flow front was locally significant, and was sometimes even
   the dominant force. An initial, short-lasting wave of high speed related to the closest of the multiple sources was observed
   in both PF and VS. The uncertainty in height and speed was generally higher in VS than in PF, in spite of the higher
   dimensionality of the second.

- We constructed partial solutions to the inverse problem, conditioning the specialized experimental design to be consistent
   with subsets of the observed data. We described the corresponding inputs sets, and investigated their intersection. We
   found model selection to be inherently linked to the inversion problem. That is, the partial inverse problems enabled us
   to select models depending on the example characteristics and spatial location.

   In particular, when attempting to correctly represent the deposits, MC performed well about 2 km upstream from the
village, while VS did so in the village. In the evaluation of the maximum flow or run-up depth, both PF and VS replicated
   the values 2 km upstream from Atenquique, but only VS replicated the values in the village. In terms of maximum flow
   speed, both PF and VS performed moderately well in the village, but only VS performed well 1 km downstream. These
   results are consistent with evolution of flow rheology downstream in the vicinity of the village, from MC above the



village, to either PF or VS within and downstream from the village. If VS was dominant as the flow propagated downstream, the meaning here may reflect an evolution from inertial to macroviscous debris flow behavior near Atenquique, perhaps related to engulfment of the reservoir just upstream from the village.

The connection of inverse problems and model uncertainty represents a fundamental challenge in the future development of multi-model solvers, suggesting as it does the advantages of dynamically selecting the model based on performance against local data for previous, example flows and their deposits.

*Code and data availability.* Datasets are available from references and Supporting Information. The $4^{\text{th}}$ release of TITAN2D is available from vhub.org.

## Appendix A: Overview of the depth-averaged models

Many models based on different assumptions from those adopted in this study are available in literature, either more complex (Pitman and Le, 2005; Iverson and George, 2014) or more simple (Dade and Huppert, 1998). We decided to focus on these three because of their historical relevance, and because they are all incorporated in our large scale mass flow simulation framework TITAN2D. We also remark that the poorly constrained data available for the past flow would make the application of a more complex model increasingly difficult.

### A1  Mohr-Coulomb

Based on the long history of studies in soil mechanics (Rankine, 1857; Drucker and Prager, 1952), the Mohr-Coulomb rheology (MC) was developed and used to represent the behavior of geophysical mass flows (Savage and Hutter, 1989). Shear and normal stress are assumed to obey Coulomb friction equation, both within the flow and at its boundaries. In other words,

$$\tau = \sigma \tan \phi, \tag{A1}$$

where $\tau$ and $\sigma$ are respectively the shear and normal stresses on failure surfaces, and $\phi$ is a friction angle. This relationship does not depend on the flow speed.

We can summarize the MC rheology assumptions as:

- *Basal Friction* based on a constant friction angle.

- *Internal Friction* based on a constant friction angle.

- *Earth pressure coefficient* formula depends on the Mohr circle (implicitly depends on the friction angles).

- Velocity based *curvature effects* are included into the equations.

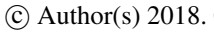



Under the assumption of symmetry of the stress tensor with respect to the $z$ axis, the earth pressure coefficient $k = k_{ap}$ can take on only one of three values $\{0, \pm 1\}$. The material yield criterion is represented by the two straight lines at angles $\pm \phi$ (the internal friction angle) relative to horizontal direction. Similarly, the normal and shear stress at the bed are represented by the line $\tau = -\sigma \tan(\delta)$ where $\delta$ is the bed friction angle.

## A1.1 MC equations

As a result, we can write down the source terms of the Eqs. (1):

$$S_x = \quad g_x h - \frac{\bar{u}}{\|\bar{\underset{\sim}{u}}\|} \left[ h \left( g_z + \frac{\bar{u}^2}{r_x} \right) \tan(\phi_{bed}) \right] - h k_{ap} \, \mathrm{sgn}\left( \frac{\partial \bar{u}}{\partial y} \right) \frac{\partial (g_z h)}{\partial y} \sin(\phi_{int})$$

$$S_y = \quad g_y h - \frac{\bar{v}}{\|\bar{\underset{\sim}{u}}\|} \left[ h \left( g_z + \frac{\bar{v}^2}{r_y} \right) \tan(\phi_{bed}) \right] - h k_{ap} \, \mathrm{sgn}\left( \frac{\partial \bar{v}}{\partial x} \right) \frac{\partial (g_z h)}{\partial x} \sin(\phi_{int}) \qquad \text{(A2)}$$

Where, $\bar{\underset{\sim}{u}} = (\bar{u}, \bar{v})$, is the depth-averaged velocity vector, $r_x$ and $r_y$ denote the radii of curvature of the local basal surface. The inverse of the radii of curvature is usually approximated with the partial derivatives of the basal slope, e.g., $1/r_x = \partial \theta_x / \partial x$, where $\theta_x$ is the local bed slope.

## A2 Pouliquen-Forterre

The scaling properties for granular flows down rough inclined planes led to the development of the Pouliquen-Forterre rheology (PF), assuming a variable frictional behavior as a function of Froude Number and flow depth (Pouliquen, 1999; Forterre and Pouliquen, 2002; Pouliquen and Forterre, 2002; Forterre and Pouliquen, 2003).

PF rheology assumptions can be summarized as:

- *Basal Friction* is based on an interpolation of two different friction angles, based on the flow regime and depth.

- *Internal Friction* is neglected.

- *Earth pressure coefficient* is equal to one.

- Normal stress is modified by a *pressure force* related to the flow thickness gradient.

- Velocity based *curvature effects* are included into the equations.

Two critical slope inclination angles are defined as functions of the flow thickness, namely $\phi_{start}(h)$ and $\phi_{stop}(h)$. The function $\phi_{stop}(h)$ gives the slope angle at which a steady uniform flow leaves a deposit of thickness $h$, while $\phi_{start}(h)$ is the angle at which a layer of thickness $h$ is mobilized. They define two different basal friction coefficients.

$$\mu_{start}(h) = \tan(\phi_{start}(h)) \qquad \text{(A3)}$$

$$\mu_{stop}(h) = \tan(\phi_{stop}(h)) \qquad \text{(A4)}$$

An empirical friction law $\mu_b(\|\bar{\underset{\sim}{u}}\|, h)$ is then defined in the whole range of velocity and thickness. The expression changes depending on two flow regimes, according to a parameter $\beta$ and the Froude number $Fr = \|\bar{\underset{\sim}{u}}\| / \sqrt{h g_z}$.





### A2.1 Dynamic friction regime - $Fr \geq \beta$

$$\mu(h, Fr) = \mu_{stop}(h\beta/Fr) \tag{A5}$$

### A2.2 Intermediate friction regime - $0 \leq Fr < \beta$

$$\mu(h, Fr) = \left(\frac{Fr}{\beta}\right)^{\gamma}\left[\mu_{stop}(h) - \mu_{start}(h)\right] + \mu_{start}(h), \tag{A6}$$

where $\gamma$ is the power of extrapolation, assumed equal to $10^{-3}$ in the sequel (Pouliquen and Forterre, 2002).

The functions $\mu_{stop}$ and $\mu_{start}$ are defined by:

$$\mu_{stop}(h) = \tan\phi_1 + \frac{\tan\phi_2 - \tan\phi_1}{1 + h/\mathcal{L}} \tag{A7}$$

and

$$\mu_{start}(h) = \tan\phi_3 + \frac{\tan\phi_2 - \tan\phi_1}{1 + h/\mathcal{L}} \tag{A8}$$

The critical angles $\phi_1$, $\phi_2$ and $\phi_3$ and the parameters $\mathcal{L}, \beta$ are the parameters of the model.

In particular, $\mathcal{L}$ is the characteristic depth of the flow over which a transition between the angles $\phi_1$ to $\phi_2$ occurs, in the $\mu_{stop}$ formula. In practice, if $h \ll \mathcal{L}$, then $\mu_{stop}(h) \approx \tan\phi_2$, and if $h \gg \mathcal{L}$, then $\mu_{stop}(h) \approx \tan\phi_1$.

### A2.3 PF equations

The depth-averaged Eqs. (1) source terms thus take the following form:

$$
\begin{aligned}
S_x &= g_x h - \frac{\bar{u}}{\|\underset{\sim}{\bar{\mathbf{u}}}\|}\left[h\left(g_z + \frac{\bar{u}^2}{r_x}\right)\mu_b(\|\underset{\sim}{\bar{\mathbf{u}}}\|, h)\right] + g_z h \frac{\partial h}{\partial x} \\
S_y &= g_y h - \frac{\bar{v}}{\|\underset{\sim}{\bar{\mathbf{u}}}\|}\left[h\left(g_z + \frac{\bar{v}^2}{r_y}\right)\mu_b(\|\underset{\sim}{\bar{\mathbf{u}}}\|, h)\right] + g_z h \frac{\partial h}{\partial y}
\end{aligned}
\tag{A9}
$$

### A3 Voellmy-Salm

The theoretical analysis of dense snow avalanches led to the VS rheology (VS) (Voellmy, 1955; Salm et al., 1990; Salm, 1993; Bartelt et al., 1999). Dense snow or debris avalanches consist of mobilized, rapidly flowing ice-snow mixed to debris-rock granules (Bartelt and McArdell, 2009). The VS rheology assumes a velocity dependent resisting term in addition to the traditional basal friction, ideally capable of including an approximation of the turbulence-generated dissipation. Many experimental and theoretical studies were developed in this framework (Gruber and Bartelt, 2007; Kern et al., 2009; Christen et al., 2010; Fischer et al., 2012). The following relation between shear and normal stresses holds:

$$\tau = \mu\sigma + \frac{\rho\|\mathbf{g}\|}{\xi}\|\underset{\sim}{\bar{\mathbf{u}}}\|^2, \tag{A10}$$

where, $\sigma$ denotes the normal stress at the bottom of the fluid layer and $\underline{\mathbf{g}} = (g_x, g_y, g_z)$ represents the gravity vector. The two parameters of the model are the bed friction coefficient $\mu$ and the velocity dependent friction coefficient $\xi$.

We can summarize VS rheology assumptions as:





    – *Basal Friction* is based on a constant coefficient, similarly to the MC rheology.

    – *Internal Friction* is neglected.

    – *Earth pressure coefficient* is equal to one.

    – Additional *turbulent friction* is based on the local velocity by a quadratic expression.

– Velocity based *curvature effects* are included into the equations, following an alternative formulation.

The effect of the topographic local curvatures is addressed with terms containing the local radii of curvature $r_x$ and $r_y$. In this case the expression is based on the speed instead of the scalar components of velocity (Pudasaini and Hutter, 2003; Fischer et al., 2012).

### A3.1 VS equations

Therefore, the final source terms take the following form:

$$
\begin{aligned}
S_x &= g_x h - \frac{\bar{u}}{\|\underset{\sim}{\bar{\mathbf{u}}}\|} \left[ h \left( g_z + \frac{\|\underset{\sim}{\bar{\mathbf{u}}}\|^2}{r_x} \right) \mu + \frac{\|\underset{\sim}{\mathbf{g}}\|}{\xi} \|\underset{\sim}{\bar{\mathbf{u}}}\|^2 \right], \\
S_y &= g_y h - \frac{\bar{v}}{\|\underset{\sim}{\bar{\mathbf{u}}}\|} \left[ h \left( g_z + \frac{\|\underset{\sim}{\bar{\mathbf{u}}}\|^2}{r_y} \right) \mu + \frac{\|\underset{\sim}{\mathbf{g}}\|}{\xi} \|\underset{\sim}{\bar{\mathbf{u}}}\|^2 \right].
\end{aligned}
\tag{A11}
$$

### Appendix B: Contributing variables

Let $[F_n(\underline{\mathbf{x}}, t)]_{n=1,\ldots,N}$ be an array of force terms, where $\underline{\mathbf{x}} \in \mathbf{R}^d$ is a spatial location, and $t \in T$ is a time instant. The degree

of contribution of those force terms to the flow dynamics can be significantly variable in space and time, and we define the *dominance factors* $[p_n(\underline{\mathbf{x}}, t)]_{n=1,\ldots,N}$, i.e., the probability of each $F_n(\underline{\mathbf{x}}, t)$ to be the dominant force. Those probabilities provide insight into the dominance of a particular source or dissipation term on the model dynamics. We remark that we focus on the modulus of the forces and hence we cope with scalar terms. It is also important to remark that all the forces depend on the input variables, and they can be thus considered as random variables. Furthermore, these definitions are general and could be

applied to any set of contributing variables, and not only to the force terms. More details about this can be found in Patra et al. (2018b).

### B1 Dominance factors

Let $(F_i)_{i \in I}$ be random variables on $(\Omega, \mathcal{F}, P_M)$. Then, $\forall i$, the dominant variable is defined as:

$$
\Phi := \max_i |F_i|.
$$



In particular, for each $j \in I$, the dominance factors are defined as:

$$p_j := P_M \left\{ \Phi = |F_j| \right\}.$$

Moreover, we define the *random contributions*, an additional tool that we use to compare the different force terms, following a less restrictive approach than the dominance factors. They are obtained dividing the force terms by the dominant force $\Phi$, and hence belong to $[0, 1]$.

## B2  Expected contributions

Let $(F_i)_{i \in I}$ be random variables on $(\Omega, \mathcal{F}, P_M)$. Then, $\forall i$, the random contribution is defined as:

$$C_i := \begin{cases} \frac{F_i}{\Phi}, & \text{if } \Phi \neq 0; \\ 0, & \text{otherwise.} \end{cases}$$

where $\Phi$ is the dominant variable. Thus, $\forall i$, the expected contributions are defined by $E[C_i]$.

In particular, for a particular location $x$, time $t$, and parameter sample $\omega$, we have $C_n(\underline{\mathbf{x}}, t, \omega) = 0$ if there is no flow or all the forces are null. The expectation of $C_n$ is reduced by the chance of $F_n$ being small compared to the other terms, or by the chance of having no flow in $(\underline{\mathbf{x}}, t)$.

*Author contributions.*  AB and AP conceived the main conceptual ideas. AB implemented and performed the simulations and the statistical
analysis. AB wrote the manuscript. AB, AP and MIB interpreted the computational results. All authors discussed the results, commented on the manuscript, provided critical feedback, and gave final approval for publication.

*Competing interests.*  The authors declare that they have no conflict of interest.

*Acknowledgements.*  We would like to acknowledge the support of NSF awards 1339765, 1521855, 1621853 and 1821311. We would like to thank Byron Rupp for his fundamental work on the localization and volume constraints of the Atenquique debris flow (Rupp, 2004), and Ali
Akhavan Safaei for the C++ and Python scripts to collect the local outputs and contributing variables on the grid elements (Akhavan-Safaei, 2018).





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
