# Peer review of "Probabilistic forecasting of plausible debris flows from Nevado de Colima (México) using data from the Atenquique debris flow, 1955"

_Natural Hazards and Earth System Sciences, 2018_

## Referee Comment (RC1) · Anonymous Referee #1 · 14 Dec 2018

The paper introduces an innovative procedure to model a debris flow through a hierarchical inversion method, with an application to the Nevado de Colima (Mexico) volcano. The paper is well written and this innovative procedure has the potential to improve our capability to model some hazards related to the volcanic activity. In my opinion this paper is certainly suitable for publication on NHESS. However, I suggest authors to address the following issues to make the paper stronger, clearer, and readable for a wider audience.

First, the authors report some statements about "falsification principle" that appears inappropriate. Popper introduces the falsification principle (which is at the core of the

scientific enterprise) considering the comparison of a model with "independent" data. This is not done here, because the comparison is made with the data that have been used to build the model. At best, this procedure can be defined as a consistency test. It is true that one model can be falsified also through such a kind of retrospective testing, but this is certainly not the golden rule to falsify a model, which, again, requires independent data. For instance, a strongly overfitting model will never be falsified using past data, but it will be certainly with future independent data. I suggest removing these statements or clarify their meaning in this paper.

Second, it is not clear to me how a model that has been calibrated inverting the data of one specific event can be used to provide a comprehensive hazard. At a first glance, this model can describe well "that" specific event; of course the model will have uncertainties that are purely epistemic (due to the fact that I do not have the real parameters and model to describe "that" event). However, this uncertainty cannot be used as an aleatory variability to describe what may happen in the future. At best in this way the authors may describe the hazard conditional to the occurrence of debris flows that have a similar source of the 1955 event. Put another way, uncertainty on the fit of one event cannot describe the variability of any kind of future event. If I misunderstood what the authors mean here, I suggest clarifying if the authors are describing "that" flow, a "similar" flow, or "any possible future" flow (as expected by a hazard model).

Third, I like very much the hierarchical structure of the inversion technique that is summarized by Figure 1. Nonetheless, the authors use a terminology which is unfamiliar to most volcanologists. To make the paper readable by a wider audience I suggest simplifying the terminology. I think that using a simpler terminology does not preclude the necessity to maintain the mathematical rigor.

Finally, as regards technical issues, the authors should double check if the symbols introduced in the paper are described in the paper when they appear. For instance, the symbol $F_0$ introduced in the point 1 of the introduction has been described only after few paragraphs.

---

## Referee Comment (RC2) · Anonymous Referee #2 · 22 Feb 2019

The manuscript presents a process to compare model outputs with real world data that focuses on understanding the relationships the model inputs have on the model outputs with respect to real world observations from Nevado de Colima volcano. The paper is well written and logically structured however it is extremely long and could be shortened. While there are no obvious grammatical issues to rectify the paper is at times difficult to read due to the reliance on what seems like a very complicated set of statistical notations. In general I think the paper should be accepted and would make an excellent contribution to the topic area. The paper would attract significant interest from a wider range of researchers struggling with the application of essentially presenting and validating predictive geophysical simulations. I only have a number

of comments or questions around the content of the paper that need to be clarified. While the approach, methods and research is very robust and accurate I think my first concern is that there is a reliance on applying models on models of models etc. with respect to analyzing the outputs of the three rheological models. It seems as though there is a missing discussion or justification of why those rheological models have those input parameters to start with. The models inputs are not arbitrarily chosen, they are incorporated or derived to represent a particular natural phenomena observed either in experimental or natural flows. While it is important to recognize the influence these input parameters have on a model outputs it seems that the original physical-numerical models that the rheological simulation was originally designed for has been over looked or ignored in favor for what I believe is a sensitivity analysis of inputs. I would also expect that more attention is paid to the rheology of the actual of Nevado de Colima (1955) flow and is better discussed with respect to the rheological model being applied or essentially outputs tested against. This brings me to my second concern is that the sites where outputs are being compared to in the real world seem static approximations of various flow characteristics whereas the comparisons from the model outputs seem to be time vary. An example is that velocity of the real world flow seems to be estimated as a single value (or a range due to the uncertainty on the outcome of the Pierson (1985) measure) yet compared to a range of velocities from the rheological model output generated over time as the simulation, simulates the flow moving past that point. Wouldn't it be better to compare the same time varying velocities between both the real flow and the simulations at the same points in time? Is it not better to look at the whole "hydro-graph" of velocities? We do know that the velocities of these types of flows do vary and pulse considerably throughout its progression. I do wonder what effect this comparisons would have on the overall technique being presented. A minor issue is that while the Pierson (1985) method is a standard method to calculate velocity from inundation it is not the most reliable measure to determine the velocity of a granular flow. Overall the paper was very interesting and makes an important point that we should move away from applying these flow simulations predictively by only calibrating

them based on passed events but present the outputs and their internal variability and uncertainties as a set of plausible outcomes. This paper should be published and will make a good contribution to the field of hazard simulations.

---

## Author Comment (AC1) · 14 Mar 2019

Dear Referee #1, attached you can find a detailed letter of response to your comments on the manuscript, as well as to the comments of the other referee. The document includes the revised manuscript, according to the letter. Best wishes, Andrea Bevilacqua, on behalf of all the authors.

Please also note the supplement to this comment:
https://www.nat-hazards-earth-syst-sci-discuss.net/nhess-2018-294/nhess-2018-294-AC1-supplement.pdf

[Figure]

[Figure]

**Supplement:**

1) The paper introduces an innovative procedure to model a debris flow through a hierarchical inversion method, with an application to the Nevado de Colima (Mexico) volcano. The paper is well written and this innovative procedure has the potential to improve our capability to model some hazards related to the volcanic activity. In my opinion this paper is certainly suitable for publication on NHESS. Thank you.

However, I suggest authors to address the following issues to make the paper stronger, clearer, and readable for a wider audience.

2) First, the authors report some statements about "falsification principle" that appears inappropriate. Popper introduces the falsification principle (which is at the core of the scientific enterprise) considering the comparison of a model with "independent" data. This is not done here, because the comparison is made with the data that have been used to build the model. At best, this procedure can be defined as a consistency test. It is true that one model can be falsified also through such a kind of retrospective testing, but this is certainly not the golden rule to falsify a model, which, again, requires independent data. For instance, a strongly overfitting model will never be falsified using past data, but it will be certainly with future independent data. I suggest removing these statements or clarify their meaning in this paper.

We appear to have slightly miscommunicated what has been done, both with respect to model development as well as the Falsification Principle. To clarify the context of our contribution with respect to the Falsification Principle, we now include a new sentence in the Introduction:

"***We remark that this statement is not related to the model selection effort, but to the model inversion effort. That is, the iterative comparison of the members of a set of disparate data to the outputs of models. We use multiple models to cover a wider span of outputs, not to seek for a best model. Indeed, we empirically falsify portions of the input spaces, not the model forms themselves.***"

Model comparison results reported in section 5.1 are not based on the Falsification Principle, and they concern specific, and expressly indicated, pieces of data. We are clarifying this in Section 1.1 (new words in bold). These data have not been used to build the models.

"…*each probability $P^j(\Omega^j_i)$ represents a performance score of the adopted model against the piece of empirical data $D_i$, and can eventually be used for model comparison purposes**, concerning that specific piece of data**.*"

We also remark that the model selection in section 6.1 is only a consequence of the inadequacy of the Mohr-Coulomb model with respect to the observational data in the village.

3) Second, it is not clear to me how a model that has been calibrated inverting the data of one specific event can be used to provide a comprehensive hazard. At a first glance, this model can describe well "that" specific event; of course the model will have uncertainties that are purely epistemic (due to the fact that I do not have the real parameters and model to describe "that" event). However, this uncertainty cannot be used as an aleatory variability to describe what may happen in the future. At best in this way the authors may describe the hazard conditional to the occurrence of debris flows

that have a similar source of the 1955 event. Put another way, uncertainty on the fit of one event cannot describe the variability of any kind of future event. If I misunderstood what the authors mean here, I suggest clarifying if the authors are describing "that" flow, a "similar" flow, or "any possible future" flow (as expected by a hazard model).

Indeed, our method gives insight into the physics of the particular flow that has been modeled. In the hazards context, such insight should first be used to inform future hazards assessment efforts regarding flows of the same type. It can be extended to assessments of hazards related to other types of flows with proper caution. We now include a new sentence in the Introduction, to better clarify the range of applicability of our method in hazard assessment problems. In this study we are not aiming to consider any possible future flow. This was already stated in the Abstract, and in the Conclusions of the manuscript.

*"We remark that our hazard assessment is related to plausible flows similar to the event that occurred in 1955. A comprehensive hazard assessment for any type of future debris flow is beyond the scope of this work, though such an approach is likely a necessary element of any comprehensive future approach using diverse data."*

4) Third, I like very much the hierarchical structure of the inversion technique that is summarized by Figure 1. Nonetheless, the authors use a terminology which is unfamiliar to most volcanologists. To make the paper readable by a wider audience I suggest simplifying the terminology. I think that using a simpler terminology does not preclude the necessity to maintain the mathematical rigor.

We have simplified our terminology by omitting the sigma-algebras, F_0, F^j_0, F^j, Fj_i. Moreover, we now include a few short sentences in section 1.2 to explain in simple words the meaning of the mathematical expressions to a more general audience.

5) Finally, as regards technical issues, the authors should double check if the symbols introduced in the paper are described in the paper when they appear. For instance, the symbol F_0 introduced in the point 1 of the introduction has been described only after few paragraphs.

Description of symbols has been checked to closely follow introduction. The sigma-algebra F_0 does not appear in the manuscript anymore.

**Anonymous Referee #2**

1) The manuscript presents a process to compare model outputs with real world data that focuses on understanding the relationships the model inputs have on the model outputs with respect to real world observations from Nevado de Colima volcano. The paper is well written and logically structured however it is extremely long and could be shortened.

To shorten the paper, we moved Figure 10 to the supplementary information, and we significantly reduced the related paragraph.

2) While there are no obvious grammatical issues to rectify the paper is at times difficult to read due to the reliance on what seems like a very complicated set of statistical notations.

Please see comment 4 and 5 to reviewer #1.

3) In general, I think the paper should be accepted and would make an excellent contribution to the topic area. The paper would attract significant interest from a wider range of researchers struggling with the application of essentially presenting and validating predictive geophysical simulations. Thank you.

I only have a number of comments or questions around the content of the paper that need to be clarified.

4) While the approach, methods and research is very robust and accurate I think my first concern is that there is a reliance on applying models on models of models etc. with respect to analyzing the outputs of the three rheological models. It seems as though there is a missing discussion or justification of why those rheological models have those input parameters to start with. The models inputs are not arbitrarily chosen; they are incorporated or derived to represent a particular natural phenomena observed either in experimental or natural flows. While it is important to recognize the influence these input parameters have on a model outputs it seems that the original physical-numerical models that the rheological simulation was originally designed for has been over looked or ignored in favor for what I believe is a sensitivity analysis of inputs.

In this study, the initial input values are not subjectively incorporated nor derived from natural flows or experiments. We present instead a procedure for the objective and replicable selection of input parameters on the basis of data. We remark that this is not equivalent to a sensitivity analysis. Indeed, the "feasible inputs" from which we start our iterative procedure are "only constrained to the existence of the numerical output and the realism of the underlying physics". A more classical discussion about the initial input choice is replaced with our new probabilistic procedure.

We included a new sentence in Section 1.1 to improve clarity. **"These initial input values are not subjectively incorporated or derived from literature or experiments."**

Moreover, we also included two new sentences in section 1.0 to better clarify why an input choice based on classical data inversion is not adopted in this study.
"**Choices based on limited data using classical inversion are often misleading since they do not reflect all potential event characteristics and can be error-prone, due to incorrectly limited event space.**

**[…]**
**In this study, we use a multi-model ensemble and a plausible region approach to provide a more prediction-oriented probabilistic framework for hazard analysis."**

The choice of the initial input values could also be built from literature. However, in our view, the feasible input space must be as wide as possible, to ensure that the procedure does not exclude any plausible output a priori, flawing the subsequent analysis.

5) I would also expect that more attention is paid to the rheology of the actual of Nevado de Colima (1955) flow and is better discussed with respect to the rheological model being applied or essentially outputs tested against.

It is true that knowledge of the actual rheology of the 1955 flow would be preferable, however, such is lacking. The procedure we outline is designed, in part, for such cases, and lends insight into what the rheology plausibly could have been, given available observations. We added several new sentences to the last paragraph in Section 2.1 to better clarify what we know a priori about the rheology of the actual flow in 1955, and how this may be reflected in the model. We remark that in this study we are not prescribing the physics of the flow based on subjective expert opinion *a priori*, but we want the data to tell us which model and input parameters may be appropriate, and which models and inputs are not. New text is marked in bold.

*"The main flood probably formed in the Atenquique ravine, but was enhanced by the confluence of flows from its tributaries: Dos Volcanes at 11.2 km, Arroyo Seco and Los Plàtanos, at 22.5 km. **The following description of the flow deposits summarizes Saucedo et al. (2008). We remark that we are not going to prescribe the rheology utilized in our modeling effort, but will rather let the data guide us to plausible rheologies and inputs.***

*During the first 10-12 km (as recorded by the proximal exposures) the flow moved down steep slopes, eroding and incorporating coarse alluvium and sand. **It left a massive deposit with a polymodal grain-size distribution, with reverse grading of coarse clasts and features typical of noncohesive debris flows, suggesting a Mohr-Coulomb rheology might be appropriate.***

*Downstream, in medial exposures, the flow encountered gentler slopes, reducing its velocity and promoting deposition of part of the sediment load. **The flow left a deposit with a bimodal grain-size distribution and better sorting than before. Dilution could have produced a change in rheology, moving toward the boundary between a debris flow and a hyperconcentrated flow.***

*Just upstream of the village, below the junction with the Arroyo Seco and Los Plàtanos ravines, eyewitnesses reported peak flood levels, possibly enhanced by the engulfing of a small water reservoir. At the junction, the flow captured the fine-material load of flow from the Arroyo Seco and Plàtanos ravines, causing significant dilution and a sudden increase in the flow turbulence **and its capacity to transport coarse particles. The resulting deposit left by the flow entering Atenquique village shows again a polymodal grain-size distribution. Here, Voellmy-Salm rheology might become more appropriate than Mohr-Coulomb. However, Pouliquen-Forterre rheology might also be able to flexibly reproduce the resulting behavior.***

*Downstream from the town, the flow lost its capacity to transport large boulders, probably due to widening and the consequent fall in velocity, which **may have been** further reduced by the hydraulic roughness effects of the flow impacting buildings. **The flood definitively transformed from a debris flow to a hyperconcentrated flow.** The diluted flow probably had a peak velocity in the range of 4 to 6 m/s, obtained using the methods of (Pierson, 1985; Saucedo et al., 2008). The flood finally continued downstream to join with the*

*perennial Tuxpan River, where it **eventually transformed to a sediment-laden flow. It** emplaced up to 6 m of deposits."*

6) This brings me to my second concern is that the sites where outputs are being compared to in the real world seem static approximations of various flow characteristics whereas the comparisons from the model outputs seem to be time vary. An example is that velocity of the real world flow seems to be estimated as a single value (or a range due to the uncertainty on the outcome of the Pierson (1985) measure) yet compared to a range of velocities from the rheological model output generated over time as the simulation, simulates the flow moving past that point. Wouldn't it be better to compare the same time varying velocities between both the real flow and the simulations at the same points in time? Is it not better to look at the whole "hydro-graph" of velocities? We do know that the velocities of these types of flows do vary and pulse considerably throughout its progression. I do wonder what effect this comparison would have on the overall technique being presented. We are comparing the estimated peak speed from the superelevation method of Pierson et al. (1985) to the maximum speed simulated. We are not comparing time varying simulated speeds with time fixed data. We are clarifying this in the text and the figure captions, by specifying that we are always comparing **peak speed at a point**.

We also added a paragraph in Section 5.1 to clarify why we are not using hydro-graphs.
**"Unfortunately, the speed of the 1955 Atenquique flow is unknown. In our analysis we consider the peak speed estimated using the superelevation method of (Pierson, 1985; Saucedo et al. 2008). We remark that, although the reconstruction of hydrographs from an analogous flow may provide us with time dependent data, this would introduce additional uncertainty that is difficult to constrain."**

We also remark that a speed range based on hydrographs would be included in the range [0-max] already calculated, hence this information would not be significantly richer than using the peak speed only.

7) A minor issue is that while the Pierson (1985) method is a standard method to calculate velocity from inundation it is not the most reliable measure to determine the velocity of a granular flow. Given the data, using more accurate measures of the peak speed would not be particularly helpful, and are likely not suitable. Moreover, the method in Pierson (1985) was referenced in Saucedo et al. (2008) as providing a good approximation to the speed of the Atenquique flow. Further improvement in the reconstruction of flow speed is outside the scope of this work.

8) Overall the paper was very interesting and makes an important point that we should move away from applying these flow simulations predictively by only calibrating them based on passed events but present the outputs and their internal variability and uncertainties as a set of plausible outcomes. This paper should be published and will make a good contribution to the field of hazard simulations. Thank you.

[revised manuscript text omitted]